# BENCHMARKING MODEL-BASED REINFORCEMENT LEARNING

## ABSTRACT

Model-based reinforcement learning (MBRL) is widely seen as having the potential to be significantly more sample efficient than model-free RL. However, research in model-based RL has not been very standardized. It is fairly common for authors to experiment with self-designed environments, and there are several separate lines of research, which are sometimes closed-sourced or not reproducible. Accordingly, it is an open question how these various existing algorithms perform relative to each other. To facilitate research in MBRL, in this paper we gather a wide collection of MBRL algorithms and propose over 18 benchmarking environments specially designed for MBRL. We benchmark these algorithms with unified problem settings, including noisy environments. Beyond cataloguing performance, we explore and unify the underlying algorithmic differences across MBRL algorithms. We characterize three key research challenges for future MBRL research: the dynamics bottleneck, the planning horizon dilemma, and the early-termination dilemma. Finally, to facilitate future research on MBRL, we open-source our benchmark[1].

## 1 INTRODUCTION

Reinforcement learning (RL) algorithms are most commonly classified in two categories: model-free RL (MFRL), which directly learns a value function or a policy by interacting with the environment, and model-based RL (MBRL), which uses interactions with the environment to learn a model of it. While model-free algorithms have achieved success in areas including robotics (Lillicrap et al., 2015; Schulman et al., 2017; Heess et al., 2017; Andrychowicz et al., 2018), video-games (Mnih et al., 2013; 2016), and character animation (Peng et al., 2018), their high sample complexity limits largely their application to simulated domains. By learning a model of the environment, model-based methods learn with significantly lower sample complexity. However, learning an accurate model of the environment has proven to be a challenging problem in certain domains. Modelling errors cripple the effectiveness of these algorithms, resulting in policies that exploit the deficiencies of the models, which is known as model-bias (Deisenroth & Rasmussen, 2011). Recent approaches have been able to alleviate the model-bias problem by characterizing the uncertainty of the learned models by the means of probabilistic models and ensembles. This has enabled model-based methods to match model-free asymptotic performance in challenging domains while using much fewer samples (Kurutach et al., 2018; Chua et al., 2018; Clavera et al., 2018).

These recent advances have led to a great excitement in the field of model-based reinforcement learning. Despite the impressive results achieved, how these methods compare against each other and against standard baselines remains unclear. Reproducibility and lack of open-source code are persistent problems in RL (Henderson et al., 2018; Islam et al., 2017), which makes it difficult to compare novel algorithms against prior lines of research. In MBRL, this problem is exacerbated by the modifications made to the environments: pre-processing of the observations, modification of the reward functions, or using different episode horizons. Such lack of standardized implementations and environments in MBRL makes it difficult to quantify scientific progress.

Systematic evaluation and comparison will not only further our understanding of the strengths and weaknesses of existing algorithms, but also reveal their limitations and suggest directions for future research. Benchmarks play a crucial role in other fields of research. For instance, MFRL has benefited

---

[1]The link to the code base is invisible during reviewing process.

greatly from the introduction of benchmarking code bases and environments such as rllab (Duan et al., 2016), OpenAI Gym (Brockman et al., 2016), and DM Control Suite (Tassa et al., 2018); where the latter two have been the de facto benchmarking platforms. Besides RL, benchmarking platforms have also accelerated areas such as computer vision (Deng et al., 2009; Lin et al., 2014), machine translation (Koehn et al., 2007) and speech recognition (Panayotov et al., 2015).

In this paper, we benchmark 11 MBRL algorithms and 4 MFRL algorithms across 18 environments based on the standard OpenAI Gym (Brockman et al., 2016). The environments, designed to hold the common assumptions in model-based methods, range from simple 2D tasks, such as Cart-Pole, to complex domains that are usually not evaluated on, such as Humanoid. The benchmark is further extended by characterizing the robustness of the different methods when stochasticity in the observations and actions is introduced. Based on the empirical evaluation, we propose three main causes that stagnate the performance of model-based methods: 1) *Dynamics bottleneck*: algorithms with learned dynamics are stuck at performance local minima significantly worse than using ground-truth dynamics, i.e. the performance does not increase when more data is collected. 2) *Planning horizon dilemma*: while increasing the planning horizon provides more accurate reward estimation, it can result in performance drops due to the curse of dimensionality and modelling errors. 3) *Early termination dilemma*: early termination is commonly used in MFRL for more directed exploration, to achieve faster learning. However, similar performance gain are not yet observed in MBRL algorithms, which limits their effectiveness in complex environments.

## 2 PRELIMINARIES

We formulate all of our tasks as a discrete-time finite-horizon Markov decision process (MDP), which is defined by the tuple $(\mathcal{S}, \mathcal{A}, p, r, \rho_0, \gamma, H)$. Here, $\mathcal{S}$ denotes the state space, $\mathcal{A}$ denotes the action space, $p(s'|a, s) : \mathcal{S} \times \mathcal{A} \times \mathcal{S} \rightarrow [0, 1]$ is transition dynamics density function, $r(s, a, s') : \mathcal{S} \times \mathcal{A} \times \mathcal{S} \rightarrow \mathbb{R}$ defines the reward function, $\rho_0$ is the initial state distribution, $\gamma$ is the discount factor, and $H$ is the horizon of the problem. Contrary to standard model-free RL, we assume access to an analytic differentiable reward function. The aim of RL is to learn an optimal policy $\pi$ that maximizes the expected total reward $J(\pi) = \mathbb{E}_{\substack{a_t \sim \pi \\ s_t \sim p}}[\sum_{t=1}^{H} \gamma^t r(s_t, a_t)]$.

**Dynamics Learning:** MBRL algorithms are characterized by learning a model of the environment. After repeated interactions with the environment, the experienced transitions are stored in a dataset $\mathcal{D} = \{(s_t, a_t, s_{t+1})\}$ which is then used to learn a dynamics function $\tilde{f}_\phi$. In the case where ground-truth dynamics are deterministic, the learned dynamics function $\tilde{f}_\phi$ predicts the next state. In stochastic settings, it is common to represent the dynamics with a Gaussian distribution, i.e., $p(s_{t+1}|a_t, s_t) \sim \mathcal{N}(\mu(s_t, a_t), \Sigma(s_t, a_t))$ and the learned dynamics model corresponds to $\tilde{f}_\phi = (\tilde{\mu}_\phi(s_t, a_t), \tilde{\Sigma}_\phi(s_t, a_t))$.

## 3 ALGORITHMS

In this section, we introduce the benchmarked MBRL algorithms, which are divided into: 1) Dyna-style Algorithms, 2) Policy Search with Backpropagation through Time, and 3) Shooting Algorithms.

### 3.1 DYNA-STYLE ALGORITHMS

In the Dyna algorithm (Sutton, 1990; 1991a;b), training iterates between two steps. First, using the current policy, data is gathered from interaction with the environment and then used to learn the dynamics model. Second, the policy is improved with imagined data generated by the learned model. Dyna algorithms learn policies using model-free algorithms with rich imaginary experience without interaction with the real environment. Dyna can also be applied to tasks with image input as in world models (Ha & Schmidhuber, 2018a;b).

**Model-Ensemble Trust-Region Policy Optimization (ME-TRPO)** (Kurutach et al., 2018): Instead of using a single model, ME-TRPO uses an ensemble of neural networks to model the dynamics, which effectively combats model-bias. The ensemble $\tilde{f}_\phi = \{\tilde{f}_{\phi_1}, ..., \tilde{f}_{\phi_K}\}$ is trained using standard squared L2 loss. In the policy improvement step, the policy is updated using Trust-Region Policy

Optimization (TRPO) (Schulman et al., 2015), on experience generated by the learned dynamics models.

**Stochastic Lower Bound Optimization (SLBO)** (Luo et al., 2019): SLBO is a variant of ME-TRPO with theoretical guarantees of monotonic improvement. In practice, instead of using single-step squared L2 loss, SLBO uses a multi-step L2-norm loss to train the dynamics.

**Model-Based Meta-Policy-Optimzation (MB-MPO)** (Clavera et al., 2018): MB-MPO forgoes the reliance on accurate models by meta-learning a policy that is able to adapt to different dynamics. Similar to ME-TRPO, MB-MPO learns an ensemble of neural networks. However, each model in the ensemble is considered as a different task to meta-train (Finn et al., 2017) on. MB-MPO meta-trains a policy that quickly adapts to any of the different dynamics of the ensemble, which is more robust against model-bias.

### 3.2 Policy Search with Backpropagation through Time

Contrary to Dyna-style algorithms, where the learned dynamics models are used to provide imagined data, policy search with backpropagation through time exploits the model derivatives. Consequently, these algorithms are able to compute the analytic gradient of the RL objective with respect to the policy, and improve the policy accordingly.

**Probabilistic Inference for Learning Control (PILCO)** (Deisenroth & Rasmussen, 2011; Deisenroth et al., 2015; Kamthe & Deisenroth, 2017): In PILCO, Gaussian processes (GPs) are used to model the dynamics of the environment. The dynamics model $f_\mathcal{D}(s_t, a_t)$ is a probabilistic and nonparametric function of the collected data $\mathcal{D}$. The policy $\pi_\theta$ is trained to maximize the RL objective by computing the analytic derivatives of the objective with respect to the policy parameters $\theta$. The training process iterates between collecting data using the current policy and improving the policy. Inference in GPs does not scale in high dimensional environments, limiting its application to simpler domains.

**Iterative Linear Quadratic-Gaussian (iLQG)** (Tassa et al., 2012): In iLQG, the ground-truth dynamics are assumed to be known by the agent. The algorithm uses a quadratic approximation on the RL reward function and a linear approximation on the dynamics, converting the problem solvable by linear-quadratic regulator (LQR) (Bemporad et al., 2002). By using dynamic programming, the optimal controller for the approximated problem is a linear time-varying controller. iLQG is a model predictive control (MPC) algorithm, where re-planning is performed at each time-step.

**Guided Policy Search (GPS)** (Levine & Abbeel, 2014; Levine et al., 2015; Zhang et al., 2016; Finn et al., 2016b; Montgomery & Levine, 2016; Chebotar et al., 2017): Guided policy search essentially distills the iLQG controllers $\pi_\mathcal{G}$ into a neural network policy $\pi_\theta$ by behavioural cloning, which minimizes $\mathbb{E}[D_{\mathrm{KL}}(\pi_\mathcal{G}(\cdot|s_t)\|\pi_\theta)]$. The dynamics are modelled to be Gaussian-linear time-varying. To prevent over-confident policy improvement that deviates from the last real-world trajectory, the reward function is augmented as $\tilde{r}(s_t, a_t) = r(s_t, a_t) - \eta D_{\mathrm{KL}}(\pi_\mathcal{G}(\cdot|s_t)\|p(\cdot|s_t))$, where $p(\cdot|s_t)$ is the passive dynamics distribution from last trajectories. In this paper, we use the MD-GPS variant (Montgomery & Levine, 2016).

**Stochastic Value Gradients (SVG)** (Heess et al., 2015): SVG tackles the problem of compounding model errors by using observations from the real environment, instead of the imagined one. To accommodate mismatch between model predictions and real transitions, the dynamics models in SVG are probabilistic. The policy is improved by computing the analytic gradient of the real trajectories with respect to the policy. Re-parametrization trick is used to permit back-propagation through the stochastic sampling.

### 3.3 Shooting Algorithms

This class of algorithms provide a way to approximately solve the receding horizon problem posed in model predictive control (MPC) when dealing with non-linear dynamics and non-convex reward functions. Their popularity has increased with the use of neural networks for modelling dynamics.

**Random Shooting (RS)** (Richards, 2005; Rao, 2009): RS optimizes the action sequence $\boldsymbol{a}_{t:t+\tau}$ to maximize the expected planning reward under the learned dynamics model, i.e., $\max_{\boldsymbol{a}_{t:t+\tau}} \mathbb{E}_{s'_t \sim \tilde{f}_\phi}[\sum_{t'=t}^{t+\tau} r(s'_t, a'_t)]$. In particular, the agent generates $K$ candidate random sequences

of actions from a uniform distribution, and evaluates each candidate using the learned dynamics. The optimal action sequence is approximated as the one with the highest return. A RS agent only applies the first action from the optimal sequence and re-plans at every time-step.

**Mode-Free Model-Based (MB-MF)** (Nagabandi et al., 2017): Generally, random shooting has worse asymptotic performance when compared with model-free algorithms. In MB-MF, the authors first train a RS controller $\pi_{RS}$, and then distill the controller into a neural network policy $\pi_\theta$ using DAgger (Ross et al., 2011), which minimizes $D_{KL}(\pi_\theta(s_t), \pi_{RS})$. After the policy distillation step, the policy is fine-tuned using standard model-free algorithms. In particular the authors use TRPO (Schulman et al., 2015).

**Probabilistic Ensembles with Trajectory Sampling (PETS-RS and PETS-CEM)** (Chua et al., 2018): In the PETS algorithm, the dynamics are modelled by an ensemble of probabilistic neural networks models, which captures both epistemic uncertainty from limited data and network capacity, and aleatoric uncertainty from the stochasticity of the ground-truth dynamics. PETS-RS is the same as RS except for different modeling of the dynamics. In PETS-CEM, the online optimization problem is solved using cross-entropy method (CEM) (De Boer et al., 2005; Botev et al., 2013) to obtain a better solution. PETS-CEM can also plan in latent space of image observations (Hafner et al., 2018).

### 3.4 MODEL-FREE BASELINES

In our benchmark, we include MFRL baselines to quantify the sample complexity and asymptotic performance gap between MFRL and MBRL. Specifically, we compare against representative MFRL algorithms including Trust-Region Policy Optimization (TRPO) (Schulman et al., 2015), Proximal-Policy Optimization (PPO) (Schulman et al., 2017; Heess et al., 2017), Twin Delayed Deep Deterministic Policy Gradient (TD3) (Fujimoto et al., 2018), and Soft Actor-Critic (SAC) (Haarnoja et al., 2018). The former two are state-of-the-art on-policy MFRL algorithms, and the latter two are considered the state-of-the-art off-policy MFRL algorithms.

## 4 EXPERIMENTS

In this section, we present the results of our benchmarking and examine the causes that stagnate the performance of MBRL methods. Specifically, we designed the benchmark to answer the following questions: 1) How do existing MBRL approaches compare against each other and against MFRL methods across environments with different complexity (Section 4.3)? 2) Are MBRL algorithms robust against observation and action noise (Section 4.4)? and 3) What are the main bottlenecks in the MBRL methods?

Aiming to answer the last question, we present three phenomena inherent of MBRL methods, which we refer to as dynamics bottleneck (Section 4.5), planning horizon dilemma (Section 4.6), and early termination dilemma (Section 4.7).

### 4.1 BENCHMARKING ENVIRONMENTS

Our benchmark consists of 18 environments with continuous state and action space based on OpenAI Gym (Brockman et al., 2016). We include a full spectrum of environments with different difficulty and episode length, from CartPole to Humanoid. More specifically, we have the following modifications:

- To accommodate traditional MBRL algorithms such as iLQG and GPS, we modify the reward function so that the gradient with respect to observation always exists or can be approximated.

- We note that early termination has not been applied in MBRL, and we specifically have both the raw environments and the variants with early termination, indicated by the suffix ET.

- The original Swimmer-v0 in OpenAI Gym was unsolvable for all algorithms. Therefore, we modified the position of the velocity sensor so that it's easier to solve. We name this easier version as Swimmer while still keep the original one as a reference, named as Swimmer-v0.

For a detailed description of the environments and the reward functions used, we refer readers to Appendix A. We also provide open-sourced version of the tasks based on Roboschool or Pybullet (AMD, 2014; Ellenberger, 2018; Klimov & Schulman, 2017), which we refer to Appendix H.

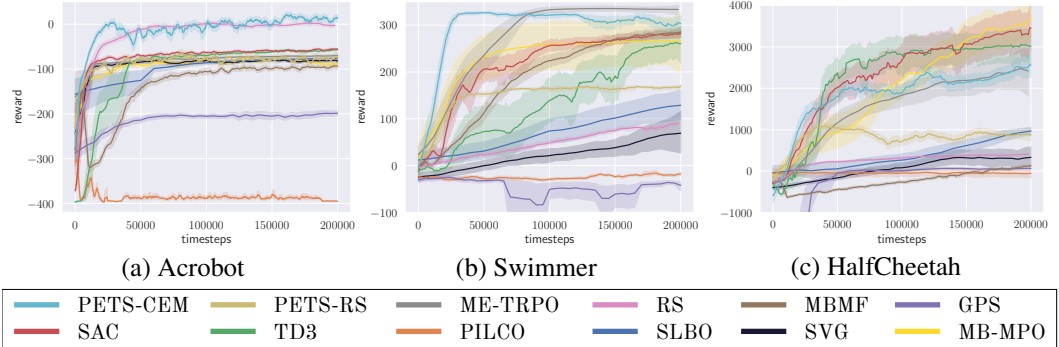

Figure 1: A subset of all 18 performance curve figures of the bench-marked algorithms. All the algorithms are run for 200k time-steps and with 4 random seeds. The remaining figures are in appendix C.

## 4.2 EXPERIMENT SETUP

**Performance Metric and Hyper-parameter Search**: Each algorithm is run with 4 random seeds. In the learning curves, the performance is averaged with a sliding window of 5 algorithm iterations. The error bars were plotted by the default Seaborn (Waskom, 2010) smoothing scheme from the mean and standard deviation of the results. Similarly, in the tables, we show the performance averaged across different random seeds with a window size of 5000 time-steps. We perform a grid search for each algorithm separately, which is summarized in appendix B. For each algorithm, We show the results using the hyper-parameters producing the best average performance.

**Training Time**: In MFRL, 1 million time-step training is common, but for many environments, MBRL algorithms converge much earlier than 200k time-steps and it takes an impractically long time to train for 1 million time-steps for some of the MBRL algorithms. We therefore show both the performance of 200k time-step training for all algorithms and show the performance of 1M time-step training for algorithms where computation is not a major bottleneck.

## 4.3 BENCHMARKING PERFORMANCE

In Table 1, we summarize the performance of each algorithm trained with 200,000 time-steps. We also include some representative performance curves in Figure 1. The learning curves for all the environments can be seen in appendix C. The engineering statistics shown in Table 2 include the computational resources, the estimated wall-clock time, and whether the algorithm is fast enough to run at real-time at test time, namely, if the action selection can be done faster than the default time-step of the environment. In Table 5, we summarize the performance ranking.

**Shooting Algorithms**: RS is very effective on simple tasks such as InvertedPendulum, CartPole and Acrobot, but as task difficulty increases RS gradually gets surpassed by PETS-RS and PETS-CEM, which indicates that modelling uncertainty aware dynamics is crucial for the performance. At the same time, PETS-CEM is better than PETS-RS in most of the environments, showing the importance of an effective planning module. However, PETS-CEM search is not as effective as PETS-RS in Ant, Walker2D and SlimHumanoid, indicating that we need more expressive and general planning module for more complex environments. MB-MF does not have obvious gains compared to other shooting algorithms, but like other model-free controllers, MB-MF can jump out of performance local-minima in MountainCar. Shooting algorithms are effective and robust across different environments.

**Dyna-Style Algorithms**: MB-MPO surpasses the performance of ME-TRPO in most of the environments and achieves the best performance in domains like HalfCheetah. Both algorithms seems to perform the best when the horizon is short. SLBO can solve MountainCar and Reacher very efficiently, but more interestingly in complex environment it achieves better performance than ME-TRPO and MB-MPO, except for in SlimHumanoid. This category of algorithms is not efficient to solve long horizon complex domains due to the compounding error effect.

**SVG**: For the majority of the tasks, SVG does not have the best sample efficiency. But for Humanoid environments, SVG is very effective compared with other MBRL algorithms. Complex environments exacerbate compounding errors; SVG which uses real observations and a value function to look into future returns, is able to surpass other MBRL algorithms in these high-dimensional domains.

Table 1: Final performance for 18 environments of the bench-marked algorithms. All the algorithms are run for 200k time-steps. Blue refers to the best methods using ground truth dynamics, red to the best MBRL algorithms, and green to the best MFRL algorithms. The results show the mean and standard deviation averaged over 4 random seeds and a window size of 5000 times-steps. "GT" indicates the model is using the ground-truth dynamics.

| | Pendulum | InvertedPendulum | Acrobot | CartPole | Mountain Car | Reacher |
|---|---|---|---|---|---|---|
| Random | -202.6 ± 249.3 | -205.1 ± 13.6 | -374.5 ± 17.1 | 38.4 ± 32.5 | -105.1 ± 1.8 | -45.7 ± 4.8 |
| ILQG | **160.8 ± 29.8** | **-0.0 ± 0.0** | -195.5 ± 28.7 | **199.3 ± 0.6** | **-55.9 ± 8.3** | **-6.0 ± 2.6** |
| GT-CEM | **170.5 ± 35.2** | **-0.2 ± 0.1** | **13.9 ± 40.5** | **199.9 ± 0.1** | **-58.0 ± 2.9** | **-3.6 ± 1.2** |
| GT-RS | **171.5 ± 31.8** | **-0.0 ± 0.0** | 2.5 ± 39.4 | **200.0 ± 0.0** | -68.5 ± 2.2 | -25.7 ± 3.5 |
| RS | 164.4 ± 9.1 | **-0.0 ± 0.0⋆** | **-4.9 ± 5.4** | **200.0 ± 0.0⋆** | -71.3 ± 0.5 | -27.1 ± 0.6 |
| MB-MF | 157.5 ± 13.2 | -182.3 ± 24.4 | -92.5 ± 15.8 | **199.7 ± 1.2** | 4.2 ± 18.5 | -15.1 ± 1.7 |
| PETS-CEM | 167.4 ± 53.0 | -20.5 ± 28.9 | **12.5 ± 29.0⋆** | **199.5 ± 3.0** | -57.9 ± 3.6 | -12.3 ± 5.2 |
| PETS-RS | 167.9 ± 35.8 | -12.1 ± 25.1 | -71.5 ± 44.6 | 195.0 ± 28.0 | -78.5 ± 2.1 | -40.1 ± 6.9 |
| ME-TRPO | **177.3 ± 1.9⋆** | -126.2 ± 86.6 | -68.1 ± 6.7 | 160.1 ± 69.1 | -42.5 ± 26.6 | -13.4 ± 0.2 |
| GPS | 162.7 ± 7.6 | -74.6 ± 97.8 | -193.3 ± 11.7 | 14.4 ± 18.6 | -10.6 ± 32.1 | -19.8 ± 0.9 |
| PILCO | 166.1 ± 23.0 | **-1.5 ± 1.6** | -394.4 ± 1.4 | **196.1 ± 13.3** | -59.0 ± 4.6 | -13.2 ± 5.9 |
| SVG | 141.4 ± 62.4 | -183.1 ± 9.0 | -79.7 ± 6.6 | 82.1 ± 31.9 | -27.6 ± 32.6 | -11.0 ± 1.0 |
| MB-MPO | 171.2 ± 26.9 | **-0.0 ± 0.0⋆** | -87.8 ± 12.9 | **199.3 ± 2.3** | -30.6 ± 34.8 | **-5.6 ± 0.8** |
| SLBO | 173.5 ± 2.5 | -240.4 ± 7.2 | -75.6 ± 8.8 | 78.0 ± 166.6 | **44.1 ± 6.8** | **-4.1 ± 0.1⋆** |
| PPO | **163.4 ± 8.0** | -40.8 ± 21.0 | -95.3 ± 8.9 | 86.5 ± 7.8 | 21.7 ± 13.1 | -17.2 ± 0.9 |
| TRPO | **166.7 ± 7.3** | -27.6 ± 15.8 | -147.5 ± 12.3 | 47.3 ± 15.7 | -37.2 ± 16.4 | -10.1 ± 0.6 |
| TD3 | **161.4 ± 14.4** | -224.5 ± 0.4 | -64.3 ± 6.9 | 196.0 ± 3.1 | -60.0 ± 1.2 | -14.0 ± 0.9 |
| SAC | **168.2 ± 9.5** | **-0.2 ± 0.1** | **-52.9 ± 2.0** | **199.4 ± 0.4** | **52.6 ± 0.6⋆** | **-6.4 ± 0.5** |

| | HalfCheetah | Swimmer-v0 | Swimmer | Ant | Ant-ET | Walker2D |
|---|---|---|---|---|---|---|
| Random | -288.3 ± 65.8 | 1.2 ± 11.2 | -9.5 ± 11.6 | 473.8 ± 40.8 | 124.6 ± 145.0 | -2456.9 ± 345.3 |
| iLQG | **2142.6 ± 137.7** | 47.8 ± 2.4 | 306.7 ± 0.8 | 9739.8 ± 745.0 | 1506.2 ± 459.4 | -1186.2 ± 126.3 |
| GT-CEM | **14777.2 ± 13964.2** | **111.0 ± 4.6** | **335.9 ± 1.1** | **12115.3 ± 209.7** | 226.0 ± 178.6 | **7719.7 ± 486.7** |
| GT-RS | 815.7 ± 38.5 | 35.8 ± 3.0 | 42.2 ± 5.3 | 2709.1 ± 631.1 | **2519.0 ± 469.8** | -1641.4 ± 137.6 |
| RS | 421.0 ± 55.2 | 31.1 ± 2.0 | 92.8 ± 8.1 | 535.5 ± 37.0 | **239.9 ± 81.7** | -2060.3 ± 228.0 |
| MB-MF | 126.9 ± 72.7 | 51.8 ± 30.9 | 284.9 ± 25.1 | 134.2 ± 50.4 | 85.7 ± 21.7 | -2218.1 ± 437.7 |
| PETS-CEM | **2795.3 ± 879.9** | 22.1 ± 25.2 | 306.3 ± 37.3 | 1165.5 ± 226.9 | 81.6 ± 145.8 | **260.2 ± 536.9** |
| PETS-RS | 966.9 ± 471.6 | 42.1 ± 20.2 | 170.1 ± 8.1 | **1852.1 ± 141.0⋆** | 130.0 ± 148.1 | **312.5 ± 493.4⋆** |
| ME-TRPO | **2283.7 ± 900.4** | 30.1 ± 9.7 | **336.3 ± 15.8⋆** | 282.2 ± 18.0 | 42.6 ± 21.1 | -1609.3 ± 657.5 |
| GPS | 52.3 ± 41.7 | 14.5 ± 5.6 | -35.3 ± 8.4 | 445.5 ± 212.9 | **275.4 ± 309.1** | -1730.8 ± 441.7 |
| PILCO | -41.9 ± 267.0 | -13.8 ± 16.1 | -18.7 ± 10.3 | 770.7 ± 153.0 | N. A. | -2693.8 ± 484.4 |
| SVG | 336.6 ± 387.6 | **77.2 ± 99.0** | 75.2 ± 85.3 | 377.9 ± 33.6 | 185.0 ± 141.6 | -1430.9 ± 230.1 |
| MB-MPO | **3639.0 ± 1185.8** | **85.0 ± 98.9⋆** | 268.5 ± 125.4 | 705.8 ± 147.2 | 30.3 ± 22.3 | -1545.9 ± 216.5 |
| SLBO | 1097.7 ± 166.4 | 41.6 ± 18.4 | 125.2 ± 93.2 | 718.1 ± 123.3 | **200.0 ± 40.1** | -1277.7 ± 427.5 |
| PPO | 17.2 ± 84.4 | **38.0 ± 1.5** | 306.8 ± 4.2 | 321.0 ± 51.2 | 80.1 ± 17.3 | -1893.6 ± 234.1 |
| TRPO | -12.0 ± 85.5 | **37.9 ± 2.0** | 215.7 ± 10.4 | 323.3 ± 24.9 | 116.8 ± 47.3 | -2286.3 ± 373.3 |
| TD3 | 3614.3 ± 82.1 | **40.4 ± 8.3** | **331.1 ± 0.9** | **956.1 ± 66.9** | 259.7 ± 1.0 | **-73.8 ± 769.0** |
| SAC | **4000.7 ± 202.1⋆** | **41.2 ± 4.6** | 309.8 ± 4.2 | 506.7 ± 165.2 | **2012.7 ± 571.3⋆** | -415.9 ± 588.1 |

| | Walker2D-ET | Hopper | Hopper-ET | SlimHumanoid | SlimHumanoid-ET | Humanoid-ET |
|---|---|---|---|---|---|---|
| Random | -2.8 ± 4.3 | -2572.7 ± 631.3 | 12.7 ± 7.8 | -1172.9 ± 757.0 | 41.8 ± 47.3 | 50.5 ± 57.1 |
| iLQG | **229.0 ± 74.7** | 1157.6 ± 224.7 | 83.4 ± 21.7 | 13225.2 ± 1344.9 | 520.0 ± 240.9 | 255.0 ± 94.6 |
| GT-CEM | **254.8 ± 233.4** | **3232.3 ± 192.3** | **256.8 ± 16.3** | **45979.8 ± 1654.9** | **1242.7 ± 676.0** | **1236.2 ± 668.0** |
| GT-RS | **207.9 ± 27.2** | -2467.2 ± 55.4 | 209.5 ± 46.8 | 8074.4 ± 441.1 | 361.5 ± 103.8 | 312.9 ± 167.8 |
| RS | 201.1 ± 10.5 | -2491.5 ± 35.1 | 247.1 ± 6.1 | -99.2 ± 388.5 | 332.8 ± 13.4 | 295.5 ± 10.9 |
| MB-MF | **350.0 ± 107.6** | -1047.4 ± 1098.7 | **926.9 ± 154.1** | -1320.2 ± 735.3 | 809.7 ± 57.5 | 776.8 ± 62.9 |
| PETS-CEM | -2.5 ± 6.8 | **1125.0 ± 679.6** | 129.3 ± 36.0 | 1472.4 ± 738.3 | 355.1 ± 157.1 | 110.8 ± 91.0 |
| PETS-RS | -0.8 ± 3.2 | -1469.8 ± 224.1 | 205.8 ± 36.5 | **2055.1 ± 771.5⋆** | 320.7 ± 182.2 | 106.9 ± 102.6 |
| ME-TRPO | -9.5 ± 4.6 | **1272.5 ± 500.9** | 4.9 ± 4.0 | -154.9 ± 534.3 | 76.1 ± 8.8 | 72.9 ± 8.9 |
| GPS | -2400.6 ± 610.8 | -768.5 ± 200.9 | -2303.9 ± 338.1 | -592.6 ± 214.1 | N. A. | N. A. |
| PILCO | N. A. | -1729.9 ± 1611.1 | N. A. | N. A. | N. A. | N. A. |
| SVG | **252.4 ± 48.4** | -877.9 ± 427.9 | 435.2 ± 163.8 | 1096.8 ± 791.0 | **1084.3 ± 77.0⋆** | 811.8 ± 241.5 |
| MB-MPO | -10.3 ± 1.4 | 333.2 ± 1189.7 | 8.3 ± 3.6 | 674.4 ± 982.2 | 115.5 ± 31.9 | 73.1 ± 23.1 |
| SLBO | 207.8 ± 108.7 | -741.7 ± 734.1 | **805.7 ± 142.4** | -588.9 ± 332.1 | 776.1 ± 252.5 | **1377.0 ± 150.4** |
| PPO | 306.1 ± 17.2 | -103.8 ± 1028.0 | 758.0 ± 62.0 | -1466.7 ± 278.5 | 454.3 ± 36.7 | 451.4 ± 39.1 |
| TRPO | 229.5 ± 27.1 | -2100.1 ± 640.6 | 237.4 ± 33.5 | -1140.9 ± 241.8 | 281.3 ± 10.9 | 289.8 ± 5.2 |
| TD3 | **3299.7 ± 1951.5⋆** | **2245.3 ± 232.4⋆** | 1057.1 ± 29.5 | **1319.1 ± 1246.1** | **1070.0 ± 168.3** | 147.7 ± 0.7 |
| SAC | **2216.4 ± 678.7** | 726.4 ± 675.5 | **1815.5 ± 655.1⋆** | **1328.4 ± 468.2** | 843.6 ± 313.1 | **1794.4 ± 458.3⋆** |

**PILCO**: PILCO achieves one of the best sample efficiency in low dimentional environments such as Cartpole, Pendulum and InvertedPendulum. But it fails in most other environments with bigger episode length and observation size, being unstable across random seeds and time-consuming to train.

**GPS**: GPS has the best performance in Ant-ET, but cannot match the best algorithms in other environments. In the original GPS, the environment is usually 100 time-step long, while most of our environments are 200 or 1000 time-step. Also GPS assumes several separate constant initial states, while our environments sample the initial state from a distribution. The deviation of trajectories between iterations can be the reason of GPS's performance drop.

| | RS | MBMF | PETS | PETS-RS | METRPO | GPS | PILCO | SVG | MB-MPO | SLBO | PPO | TRPO | TD3 | SAC |
|---|---|---|---|---|---|---|---|---|---|---|---|---|---|---|
| Reacher2D | 9.23 | 4.03 | 4.64 | 2.68 | 4.76 | 1.1 | 120 | 1.61 | 30.9 | 2.38 | 0.02 | 0.02 | 2.9 | 2.38 |
| Cheetah | 8.83 | 4.05 | 15.3 | 6.76 | 5.23 | 3.3 | N. A. | 1.41 | 57.5 | 4.96 | 0.04 | 0.02 | 4.3 | 2.21 |
| Ant | 8.2 | 5.25 | 6.5 | 5.01 | 3.46 | 5.1 | N. A. | 1.49 | 55.2 | 5.46 | 0.07 | 0.05 | 3.6 | 3.15 |
| Humanoid-ET | 13.9 | 5.05 | 7.03 | 5.1 | 5.68 | N. A. | N. A. | 1.92 | 41.4 | 5.5 | 0.05 | 0.04 | 5.37 | 3.35 |
| Slimhumanoid-ET | 9.5 | 3.3 | 4.76 | 3.35 | 2.58 | N. A. | N. A. | 1.06 | 41.5 | 6.86 | 0.03 | 0.03 | 3.13 | 4.05 |
| Slimhumanoid | 11.73 | 4.8 | 6.6 | 5.06 | 2.36 | 17.24 | N. A. | 1.05 | 41.6 | 6.8 | 0.04 | 0.03 | 3.95 | 3.15 |
| Real-time testing | ✗ | ✓ | ✗ | ✗ | ✓ | ✓ | ✓ | ✓ | ✓ | ✓ | ✓ | ✓ | ✓ | ✓ |
| CPU/GPU used | 20 / 0 | 20 / 0 | 4 / 1 | 4 / 1 | 4 / 1 | 5 / 0 | 4 / 1 | 2 / 0 | 8 / 0 | 12 / 0 | 5 / 0 | 5 / 0 | 12 / 0 | 12 / 0 |

Table 2: Wall-clock time in hours for each algorithm trained for 200k time-steps.

**MF baselines**: SAC and TD3 are two very powerful baselines with very stable performance across different environments. In general model-free and model-based methods are two almost evenly matched rivals when trained for 200,000 time-steps.

**MB with Ground-truth Dynamics**: Algorithms with ground-truth dynamics can solve the majority of the tasks, except for some of the tasks such as MountainCar. With the increasing complexity of the environments, shooting methods gradually have much better performance than the policy search methods such as iLQG, whose linear quadratic assumption is not a good approximation anymore. Early termination cause a lot of troubles for model-based algorithms, both with and without ground-truth dynamics, which is further studied in section 4.7.

## 4.4 NOISY ENVIRONMENTS

In this section, we study the robustness of each algorithm with respect to the noise added to the observation and actions. Specifically, we added Gaussian white noise to the observations and actions with standard deviation $\sigma_o$ and $\sigma_a$, respectively. In Table 3 we show the results for the HalfCheetah environment, for the full results we refer the reader to appendix D.

| HalfCheetah | iLQG | GT-PETS | RS | PETS | ME-TRPO | SVG | MB-MPO | SLBO | GT-RS | PETS-RS | SAC | TD3 |
|---|---|---|---|---|---|---|---|---|---|---|---|---|
| Original Performance | 2142.6 | 14777.2 | 421 | 2795.3 | 2283.7 | 336.6 | 3639.0 | 1097.7 | 815.7 | 966.9 | 4000.7 | 3614.3 |
| Change / $\sigma_o = 0.1$ | -2167.9 | -13138.7 | -274.8 | -915.8 | -1874.3 | -336.5 | -1282.6 | -885.2 | -809.1 | -749.9 | -2854 | -2718.6 |
| Change / $\sigma_o = 0.01$ | -1955.4 | -5550.7 | +2.1 | -385 | -886.8 | -95.8 | -3.5 | +147.1 | -322.4 | -152 | -131.5 | -2797 |
| Change / $\sigma_a = 0.1$ | -1881.4 | -3292.7 | +24.8 | -367.8 | -963.9 | -173.1 | -266.1 | +495.5 | -210.9 | 161.7 | -470.2 | 642.2 |
| Change / $\sigma_a = 0.03$ | -1832.5 | -1616.6 | +21.3 | -368.1 | -160.8 | -314.7 | +79.7 | -366.6 | -170 | 50.6 | -292.6 | 327.5 |

Table 3: The relative changes of performance of each algorithm in noisy HalfCheetah environments. We use bold text to indicates a decrease of performance >10% of the performance without noise.

As expected, adding noise is in general detrimental to the performance of the MBRL algorithms. ME-TRPO and SLBO are more likely to suffer from a catastrophic performance drop when compared to shooting methods such as PETS and RS, suggesting that re-planning successfully compensates for the uncertainty. On the other hand, the Dyna-style method MB-MPO presents to be very robust against noise. Due to the limited exploration in baseline, the performance is sometimes increased after adding noise that encourages exploration.

## 4.5 DYNAMICS BOTTLENECK

We further run MBRL algorithms for 1M time-steps on HalfCheetah, Walker2D, Hopper, and Ant environments to capture the asymptotic performance, as are shown in Table 4 and Figure 2. The results show that MBRL algorithms plateau at a performance level well below their model-free counterparts and themselves with ground-truth dynamics. This points out that when learning models, more data does not result in better performance. For instance, PETS's performance plateaus after 400k time-steps at a value much lower than the performance when using the ground-truth dynamics. We also study the performance with different network capacity, as well as using linear of RBF parameterization, as summarized in Appendix G.

The following assumptions can potentially explain the dynamics bottleneck. 1) The prediction error accumulates with time, and MBRL inevitably involves prediction on unseen states. While techniques such as probabilistic ensemble were proposed to capture uncertainty, it can be seen empirically in our paper as well as in Chua et al. (2018), that prediction becomes unstable and inaccurate with time. 2) The policy and the learning of dynamics is coupled, which makes the agents more prone to performance local-minima. While exploration and off-policy learning have been studied in Bellemare et al. (2016); Dearden et al. (1999); Wiering & Schmidhuber (1998); Houthooft et al. (2016); Schaul et al. (2019); Fujimoto et al. (2018), it has been barely addressed on current model-based approaches.

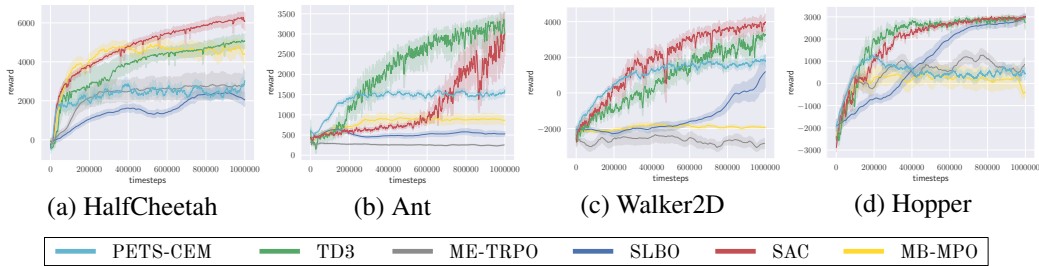

| | (a) HalfCheetah | (b) Ant | (c) Walker2D | (d) Hopper |
|---|---|---|---|---|

PETS-CEM    TD3    ME-TRPO    SLBO    SAC    MB-MPO

Figure 2: Performance curve for each algorithm trained for 1 million time-steps.

| | GT-CEM | PETS-CEM | ME-TRPO | MB-MPO | SLBO | TD3 | SAC |
|---|---|---|---|---|---|---|---|
| HalfCheetah | 14777.2 ± 13964.2 | 2875.9 ± 1132.2 | 2672.7 ± 1481.6 | 4513.1 ± 1045.4 | 2041.4 ± 932.7 | 5072.9 ± 815.8 | 6095.5 ± 936.1 |
| Walker2D | 7719.7 ± 486.7 | 1931.7 ± 667.3 | -2947.1 ± 640.0 | -1793.7 ± 80.6 | 1371.7 ± 2761.7 | 3293.6 ± 644.4 | 3941.0 ± 985.3 |
| Hopper | 3232.3 ± 192.3 | 288.4 ± 988.2 | 948.0 ± 854.3 | -495.2 ± 265.0 | 2963.1 ± 323.4 | 2745.7 ± 546.7 | 3020.3 ± 134.6 |
| Ant | 12115.3 ± 209.7 | 1675.0 ± 108.6 | 262.7 ± 36.5 | 810.8 ± 240.6 | 513.6 ± 182.0 | 3073.8 ± 773.8 | 2989.9 ± 1182.8 |

Table 4: Bench-marking performance for 1 million time-steps.

## 4.6 PLANNING HORIZON DILEMMA

One of the critical choices in shooting methods is the planning horizon. In Figure 3, we show the performance of iLQG, CEM and RS, using the same number of candidate planning sequences, but with different planning horizon. We notice that increasing the planning horizon does not necessarily increase the performance, and more often instead decreases the performance. This happens both when using ground-truth dynamics and using learned dynamics. We argue that this is result of insufficient planning in a search space which increases exponentially with planning depth, i. e., the curse of dimensionality, as is also observed in Vemula et al. (2019); Hafner et al. (2018).

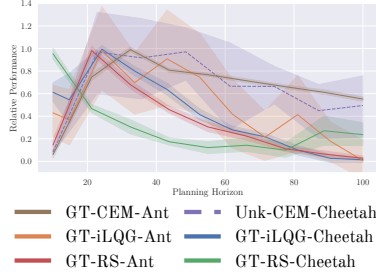

GT-CEM-Ant    Unk-CEM-Cheetah
GT-iLQG-Ant    GT-iLQG-Cheetah
GT-RS-Ant    GT-RS-Cheetah

Figure 3: The relative performance with different planning horizon.

However, in more complex environments such as the ones with early terminations, short planning horizon can lead to catastrophic performance drop, which we discuss in appendix I. We further experiment with the imaginary environment length in Dyna algorithms. We have similar results that increasing horizon does not necessarily help the performance, which is summarized in appendix F.

## 4.7 EARLY TERMINATION DILEMMA

Early termination, when the episode is finalized before the horizon has been reached, is a standard technique used in MFRL algorithms to prevent the agent from visiting unpromising states or damaging states for real robots (Peng et al., 2018; 2016; Merel et al., 2017; Heess et al., 2016; Brockman et al., 2016). When early termination is applied to the real environments, MBRL can correspondingly also apply early termination in the planned trajectories, or generate early terminated imaginary data. However, we find this technique hard to integrate into the existing MB algorithms. The results, shown in Table 1, indicates that early termination does in fact decrease the performance for MBRL algorithms of different types. We further experiment with addition schemes to incorporate early termination, summarized in appendix I. However none of them were successful. We argue that to perform efficient learning in complex environments, such as Humanoid, early termination is almost necessary. We leave it as an important request for research.

| | RS | MB-MF | PETS-CEM | PETS-RS | ME-TRPO | GPS | PILCO | SVG | MB-MPO | SLBO |
|---|---|---|---|---|---|---|---|---|---|---|
| Mean rank | 5.3 / 10 | 5.6 / 10 | 4.1 / 10 | 5 / 10 | 5.8 / 10 | 7.9 / 10 | 8.5 / 10 | 5.0 / 10 | 4.7 / 10 | 4.1 / 10 |
| Median rank | 5.5 / 10 | 7 / 10 | 4 / 10 | 5 / 10 | 6.5 / 10 | 8.5 / 10 | 10 / 10 | 4 / 10 | 4.5 / 10 | 3.5 / 10 |

Table 5: The ranking of the MBRL algorithms in the 18 benchmarking environments

## 5 CONCLUSIONS

In this paper, we benchmark the performance of a wide collection of existing MBRL algorithms, evaluating their sample efficiency, asymptotic performance and robustness. Through systematic evaluation and comparison, we characterize three key research challenges for future MBRL research. Across this very substantial benchmarking, there is no clear consistent best MBRL algorithm, suggesting lots of opportunities for future work bringing together the strengths of different approaches.

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

## A  ENVIRONMENT OVERVIEW

We provide an overview of the environments in this section. Table 6 shows the dimensionality and horizon lengths of those environments, and Table 7 specifies their reward functions.

| Environment Name | Observation Space Dimension | Action Space Dimension | Horizon |
|---|---|---|---|
| Acrobot | 6 | 1 | 200 |
| Pendulum | 3 | 1 | 200 |
| InvertedPendulum | 4 | 1 | 100 |
| CartPole | 4 | 1 | 200 |
| MountainCar | 2 | 1 | 200 |
| Reacher2D (Reacher) | 11 | 2 | 50 |
| HalfCheetah | 17 | 6 | 1000 |
| Swimmer-v0 | 8 | 2 | 1000 |
| Swimmer | 8 | 2 | 1000 |
| Hopper | 11 | 3 | 1000 |
| Ant | 28 | 8 | 1000 |
| Walker 2D | 17 | 6 | 1000 |
| Humanoid | 376 | 17 | 1000 |
| SlimHumanoid | 45 | 17 | 1000 |

Table 6: Dimensions of observation and action space, and horizon length for most of the environments used in the experiments.

| Environment Name | Reward Function $R_t$ |
|---|---|
| Acrobot | $-\cos\theta_{1,t} - \cos\left(\theta_{1,t} + \theta_{2,t}\right)$ |
| Pendulum | $-\cos\theta_t - 0.1\sin\theta_t - 0.1\dot{\theta}^2 - 0.001a_t^2$ |
| InvertedPendulum | $-\theta_t^2$ |
| CartPole | $\cos\theta_t - 0.01x_t^2$ |
| MountainCar | $position_t$ |
| Reacher2D (Reacher) | $-distance_t - \|\boldsymbol{a}_t\|_2^2$ |
| HalfCheetah | $\dot{x}_t - 0.1\|\boldsymbol{a}_t\|_2^2$ |
| Swimmer-v0 | $\dot{x}_t - 0.0001\|\boldsymbol{a}_t\|_2^2$ |
| Swimmer | $\dot{x}_t - 0.0001\|\boldsymbol{a}_t\|_2^2$ |
| Hopper | $\dot{x}_t - 0.1\|\boldsymbol{a}_t\|_2^2 - 3.0 \times (z_t - 1.3)^2$ |
| Ant | $\dot{x}_t - 0.1\|\boldsymbol{a}_t\|_2^2 - 3.0 \times (z_t - 0.57)^2$ |
| Walker2D | $\dot{x}_t - 0.1\|\boldsymbol{a}_t\|_2^2 - 3.0 \times (z_t - 1.3)^2$ |
| Humanoid | $50/3 \times \dot{x}_t - 0.1\|\boldsymbol{a}_t\|_2^2 - 5e{-}6 \times impact + 5 \times bool(1.0 <= z_t <= 2.0)$ |
| SlimHumanoid | $50/3 \times \dot{x}_t - 0.1\|\boldsymbol{a}_t\|_2^2 + 5 \times bool(1.0 <= z_t <= 2.0)$ |

Table 7: Reward function for most of the environments used in the experiments. The tasks specified by the reward functions are discussed in further detail in Section A.1.

### A.1  ENVIRONMENT-SPECIFIC DESCRIPTIONS

In this section, we provide details about environment-specific dynamics and goals.

**Acrobot**  The dynamical system consists of a pendulum with two links. The joint between the two links is actuated. Initially, both links point downwards. The goal is to swing up the pendulum, such that the tip of the pendulum reaches a given height. Let $\theta_{1,t}$, $\theta_{2,t}$ be the joint angles of the first (with one end fixed to a hinge) and second link at time $t$. The 6-dimensional observation at time $t$ is the tuple: $(\cos\theta_{1,t}, \sin\theta_{1,t}, \cos\theta_{2,t}, \sin\theta_{2,t}, \dot{\theta}_{1,t}, \dot{\theta}_{2,t})$. The reward is the height of the tip of the pendulum: $R_t = -\cos\theta_{1,t} - \cos\left(\theta_{1,t} + \theta_{2,t}\right)$.

**Pendulum**   A single-linked pendulum is fixed on the one end, with an actuator located on the joint. The goal is to keep the pendulum at the upright position. Let $\theta_t$ be the joint angle at time $t$. The 3-dimensional observation at time $t$ is $(\cos\theta_t, \sin\theta_t, \dot{\theta}_t)$ The reward penalizes the position and velocity deviation from the upright equilibrium, as well as the magnitude of the control input.

**InvertedPendulum**   The dynamical system consists of a cart that slides on a rail, and a pole connected through an unactuated joint to the cart. The only actuator applies force on the cart along the rail. The actuator force is a real number. Let $\theta_t$ be the angle of the pole away from the upright vertical position, and $x_t$ be the position of the cart away from the centre of the rail at time $t$. The 4-dimensional observation at time $t$ is $(x_t, \theta_t, \dot{x}_t, \dot{\theta}_t)$. The reward $-\theta_t^2$ penalizes the angular deviation from the upright position.

**CartPole**   The dynamical system of Cart-Pole is very similar to that of the Inverted Pendulum environment. The differences are: 1) the real-valued actuator input is discretized to $-1, 1$, with a threshold at zero; 2) the reward $\cos\theta_t - 0.01x_t^2$ indicates that the goal is to make the pole stay upright, and the cart stay at the centre of the rail.

**MountainCar**   A car is initially positioned between two "mountains", and can drive on a one-dimensional track. The goal is to reach the top of the "mountain" on the right. However, the engine of the car is not strong enough for it to drive up the valley in one go, so the solution is to drive back and forth to accumulate momentum. The observation at time $t$ is the tuple $(position_t, velocity_t)$, where both the position and velocity are one-dimensional, with respect to the track. The reward at time $t$ is simply $position_t$. Note that we use a fixed horizon, so that the agent is encouraged to reach the goal as soon as possible.

**Reacher2D (or Reacher)**   An arm with two links is fixed at one end, and is free to move on the horizontal 2D plane. There are two actuators, located at the two joints respectively. At each episode, a target is randomly placed on the 2D plane within reach of the arm. The goal is to make the tip of the arm reach the target as fast as possible, and with the smallest possible control input. Let $\boldsymbol{\theta}_t$ be the two joint positions, $\boldsymbol{x}_{target,t}$ be the position of the target, and $\boldsymbol{x}_{tip,t}$ be the position of the tip of the arm at time $t$, respectively. The observation is $(\cos\boldsymbol{\theta}_t, \sin\boldsymbol{\theta}_t, \boldsymbol{x}_{target,t}, \dot{\boldsymbol{\theta}}_t, \boldsymbol{x}_{tip,t} - \boldsymbol{x}_{target,t})$. The reward at time $t$ is $||\boldsymbol{x}_{tip,t} - \boldsymbol{x}_{target,t}||_2^2 - ||\boldsymbol{a}_t||_2^2$, where the first term is the Euclidean distance between the tip and the target.

**HalfCheetah**   Half Cheetah is a 2D robot with 7 rigid links, including 2 legs and a torso. There are 6 actuators located at 6 joints respectively. The goal is to run forward as fast as possible, while keeping control inputs small. The observation include the (angular) position and velocity of all the joints (including the root joint, whose position specifies the robot's position in the world coordinate), except for the $x$ position of the root joint. The reward is the $x$ direction velocity plus penalty for control inputs.

**Swimmer-v0**   Swimmer-v0 is a 2D robot with 3 rigid links, sliding on a 2D plane. There are 2 actuators, located on the 2 joints between the links. The root joint is located at the centre of the middle link. The observation include the (angular) position and velocity of all the joints, except for the position of the two slider joints (indicating the $x$ and $y$ positions). The reward is the $x$ direction velocity plus penalty for control inputs.

**Swimmer**   The dynamical system of Swimmer is similar to that of Swimmer-v0, except that the root joint is located at the tip of the first link (i.e. the "head" of the swimmer).

**Hopper**   Hopper is a 2D "robot leg" with 4 rigid links, including the torso, thigh, leg and foot. There are 3 actuators, located at the three joints connecting the links. The observation include the (angular) position and velocity of all the joints, except for the $x$ position of the root joint. The reward is the $x$ direction velocity plus penalty for the distance to a target height and control input. The intended goal is to hop forward as fast as possible, while approximately maintaining the standing height, and with the smallest control input possible. We also add an alive bonus of 1 to the agents at every time-step, which is also applied to Ant, Walker2D.

**Ant**   Ant is a 3D robot with 13 rigid links, including a torso 4 legs. There are 8 actuators, 2 for each leg, located at the joints. The observation include the (angular) position and velocity of all the joints, except for the $x$ and $y$ positions of the root joint. The reward is the $x$ direction velocity plus penalty for the distance to a target height and control input. The intended goal is to go forward, while approximately maintaining the normal standing height, and with the smallest control input possible.

**Walker2D**   Walker 2D is a planar robot, consisting of 7 rigid links, including a torso and 2 legs. There are 6 actuators, 3 for each leg. The observation include the (angular) position and velocity of all the joints, except for the $x$ position of the root joint. The reward is the $x$ direction velocity plus penalty for the distance to a target height and control input. The intended goal is to walk forward as fast as possible, while approximately maintaining the standing height, and with the smallest control input possible.

**Humanoid**   Humanoid is a 3D human shaped robot consisting of 13 rigid links. There are 17 actuators, located at the humanoid's abdomen, hips, knees, shoulders and elbows. The observation space include the joint (angular) positions and velocities, centre of mass based inertia, velocity, external force, and actuator force. The reward is the scaled $x$ direction velocity, plus penalty for control input, impact (external force) and undesired height.

**SlimHumanoid**   The dynamical system of Slim Humanoid is similar to that of Humanoid, except that the observation is simply the joint positions and velocities, without the center of mass based quantities, external force and actuator force. Also, the reward no longer penalizes the impact (external force).

## B   Hyper-parameter Search and Engineering Details

In this section, we provide a more detailed description of the hyper-parameters we search for each algorithm. Note that we select the best hyper-parameter combination for each algorithm, but we still provide a reference hyper-parameter combination that is generally good for all environments.

### B.1   iLQG

For iLQG algorithm, the hyper-parameters searched are summarized in 8. While the recommended hyper-parameters usually have the best performance, they can result in more computation resources needed. In the following sections, number of planning trajectory is also refereed as search population size.

| Hyper-parameter | Value Tried | Recommended Value |
|---|---|---|
| planning horizon | 10, 20, 30, 50, 100 | 20 |
| max linesearch backtrack | 1, 5, 10, 15, 20 | 10 |
| number iLQG update per time-step | 1, 5, 10, 20 | 10 |
| number of planning trajectory | 1, 2, ..., 10, 20 | 10 |

Table 8: Hyper-parameter grid search options for iLQG.

### B.2   Ground-truth CEM and Ground-truth RS

For the CEM and RS with ground-truth dynamics, we search only with different planning horizon, search population size. which include 10, 20, 30, 40, 50, 60, 70, 80, 90, 100. As also mentioned in planning horizon dilemma in section 4.6, the best planning horizon is usually 20 to 30.

### B.3   RS, PETS and PETS-RS

We mention that in Nagabandi et al. (2017), RS has very different hyper-parameter sets from the RS studied in PETS-RS Chua et al. (2018). The search of hyper-parameters for RS is the same for

| Hyper-parameter | Value Tried | Recommended Value |
|---|---|---|
| planning horizon | 10, 20, 30, ..., 90, 100 | 30 |
| search population size | 500, 1000, 2000 | 1000 |

Table 9: Hyper-parameter grid search options for RS, CEM using ground-truth dynamics.

RS using ground-truth dynamics as illustrated in the Table 9. The PETS and PETS is searched with the hyper-parameters in Table 10. For simpler environments, it is usually better to use a planning horizon of 30. For environments such as Walker2D and Hopper, 100 is the best planning horizon. We

| Hyper-parameter | Value Tried | Recommended Value |
|---|---|---|
| planning horizon | 10, 30, 50, 60, 70, 80, 90, 100 | 30 / 100 |
| search population size | 50, 100, 500, 1000, 2000 | 500 |
| elite size | 50, 100, 150 | 50 |
| PETS combination | D-E, DE-E, PE-E, PE-TSinf, PE-TS1, PE-DS | PE-E |

Table 10: Hyper-parameter grid search options for RS.

note that for the dynamics-propagation combination, we choose PE-E not because of having the best performance. PE-E is among the best models, with comparable performance to other combinations such as PE-DS, PE-TS1, PE-TSinf. However PE-E is very computation efficient compared to other variants. For example, PE-DS on PETS-CEM costs 68 hours for one random seed for HalfCheetah with planning horizon of 30 to train for 200,000 time-steps. While PE-E usually only takes about 5 hours, and is suitable for research. The best models for HalfCheetah uses planning horizon of 100, and takes about 15 hours.

We also note that with some trivial pre-processing of the observation function, such as using sin and cos functions to the angle observation, the performance can be much better for PETS. We include the performance on the pre-processed task provided in the original PETS paper Chua et al. (2018), and compare it with the performance on the provided task in the benchmark in Figure 4. We note that for the modified environments in Chua et al. (2018), the performance can reach more than 10000 for 200k time-steps.

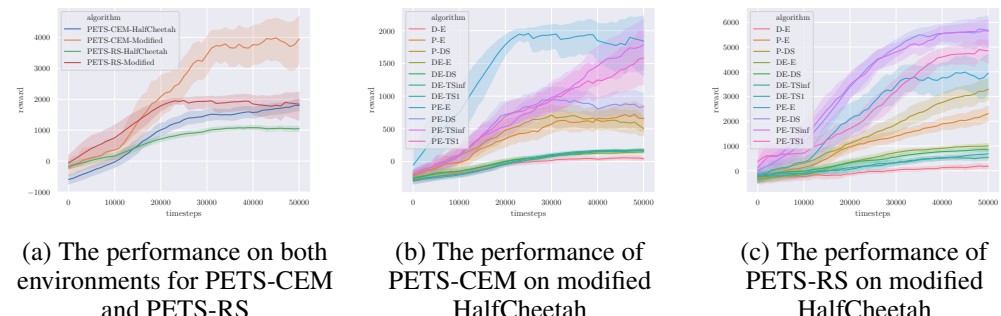

(a) The performance on both environments for PETS-CEM and PETS-RS

(b) The performance of PETS-CEM on modified HalfCheetah

(c) The performance of PETS-RS on modified HalfCheetah

Figure 4: The performance of HalfCheetah from our benchmark, and the HalfCheetah from the PETS paper Chua et al. (2018), which we refer to as "Modified". We also include the performance of PETS-CEM and PETS-RS using different dynamics-propagation combination on the modified HalfCheetah from Chua et al. (2018).

In the benchmark results in the main paper, PETS-RS uses the search scheme in Table 10, except for it does not have hyper-parameters of elite size.

## B.4 MBMF

Originally MBMF was designed to run for 1 million time-steps Nagabandi et al. (2017). Therefore, to accommodate the algorithm with 200,000 time-steps, we perform the search in Table 11.

| Hyper-parameter | Value Tried | Recommended Value |
|---|---|---|
| trust region method | TRPO, PPO | PPO |
| search population size | 1000, 5000, 2000 | 5000 |
| planning horizon | 10, 20, 30 | 20 |
| time-steps per iteration | 1000, 2000, 5000 | 1000 |
| model based time-steps | 5000, 7000, 10000 | 7000 |
| dagger epoch | 100, 300, 500 | 300 |

Table 11: Hyper-parameter grid search options for MBMF.

## B.5 METRPO AND SLBO

For METRPO and SLBO, we search for the following hyper-parameters. We note that for environments with episode length of 100 or 200, we always use the same length for imaginary episodes. We also refer to Appendix F for more details.

| Hyper-parameter | Value Tried | Recommended Value |
|---|---|---|
| imaginary episode length | 1000, 500, 200, 100 | 1000 |
| TRPO iterations | 1, 10, 20, 30, 40 | 20 / 40 |
| network ensembles | 1, 5, 10, 20 | 5 |
| Terminate imaginary episode | True, False | False |

Table 12: Hyper-parameter grid search options for METRPO and SLBO.

## B.6 GPS

The GPS is based on the code-base Finn et al. (2016a). We note that in the original code-base, the agent samples the initial state from several separate conditions. For each condition, there is not randomness of the initial state. However, in our bench-marking environments, the initial state is sample from a Gaussian distribution, which is essentially making the environments harder to solve.

| Hyper-parameter | Value Tried | Recommended Value |
|---|---|---|
| time-step per iteration | 1000, 5000, 10000 | 5000 |
| kl step | 0.5, 1.0, 2.0, 0.5 | 1.0 |
| dynamics Gaussian mixture model clusters | 5, 10, 20, 30 | 20 |
| policy Gaussian mixture model clusters | 10, 20 | 20 |

Table 13: Hyper-parameter grid search options for GPS.

## B.7 PILCO

For PILCO, we search for the following hyper-parameter in Table 14. We note that PILCO is very unstable across random seeds. Also, it is quite common for PILCO algorithms to add additional penalty in existing code-bases using human priors. We argue that it is unfair to other algorithms and we remove any additional reward functions. Also, for PILCO to train for 200,000 time-steps, we have to use a data-set to increase training efficiency.

## B.8 SVG

For SVG, we reproduce the variant of SVG-1 with experience replay, which is claimed in Heess et al. (2015).

| Hyper-parameter | Value Tried | Recommended Value |
|---|---|---|
| Optimizing Horizon | 30, 100, 200, adaptive | 100 or 30 |
| episode per iteration | 1, 2, 4 | 1 |
| data-set size | 200, 1000, 2000, 20000, 40000, 10000 | 200 or 1000 |

Table 14: Hyper-parameter grid search options for PILCO.

| Hyper-parameter | Value Tried | Recommended Value |
|---|---|---|
| SVG learning rate | 0.00003, 0.0001, 0.0003, 0.001 | 0.0001 |
| data buffer size | 25000 | 25000 |
| KL penalty | 0.0003, 0.001, 0.003 | 0.001 |

Table 15: Hyper-parameter grid search options for SVG.

## B.9 MB-MPO

In this algorithm, we use most the hyper-parameters in the original paper Clavera et al. (2018), except in the ones the algorithm is more sensitive to.

| Hyper-parameter | Value Tried | Recommended Value |
|---|---|---|
| inner learning rate | 0.0005, 0.001, 0.01 | 0.0005 |
| rollouts per task | 10, 20, 30 | 20 |
| MAML iterations | 30, 50, 75 | 50 |

Table 16: Hyper-parameter grid search options for MB-MPO.

## B.10 MODEL-FREE BASELINES

For PPO and TRPO, we search for different time-steps samples in one iteration. For SAC and TD3, we use the default values from Haarnoja et al. (2018) and Fujimoto et al. (2018) respectively.

| Hyper-parameter | Value Tried | Recommended Value |
|---|---|---|
| time-steps per iteration | 1000, 2000, 5000, 20000 | 2000 |

Table 17: Hyper-parameter grid search options for model-free algorithms.

# C  DETAILED BENCH-MARKING PERFORMANCE RESULTS

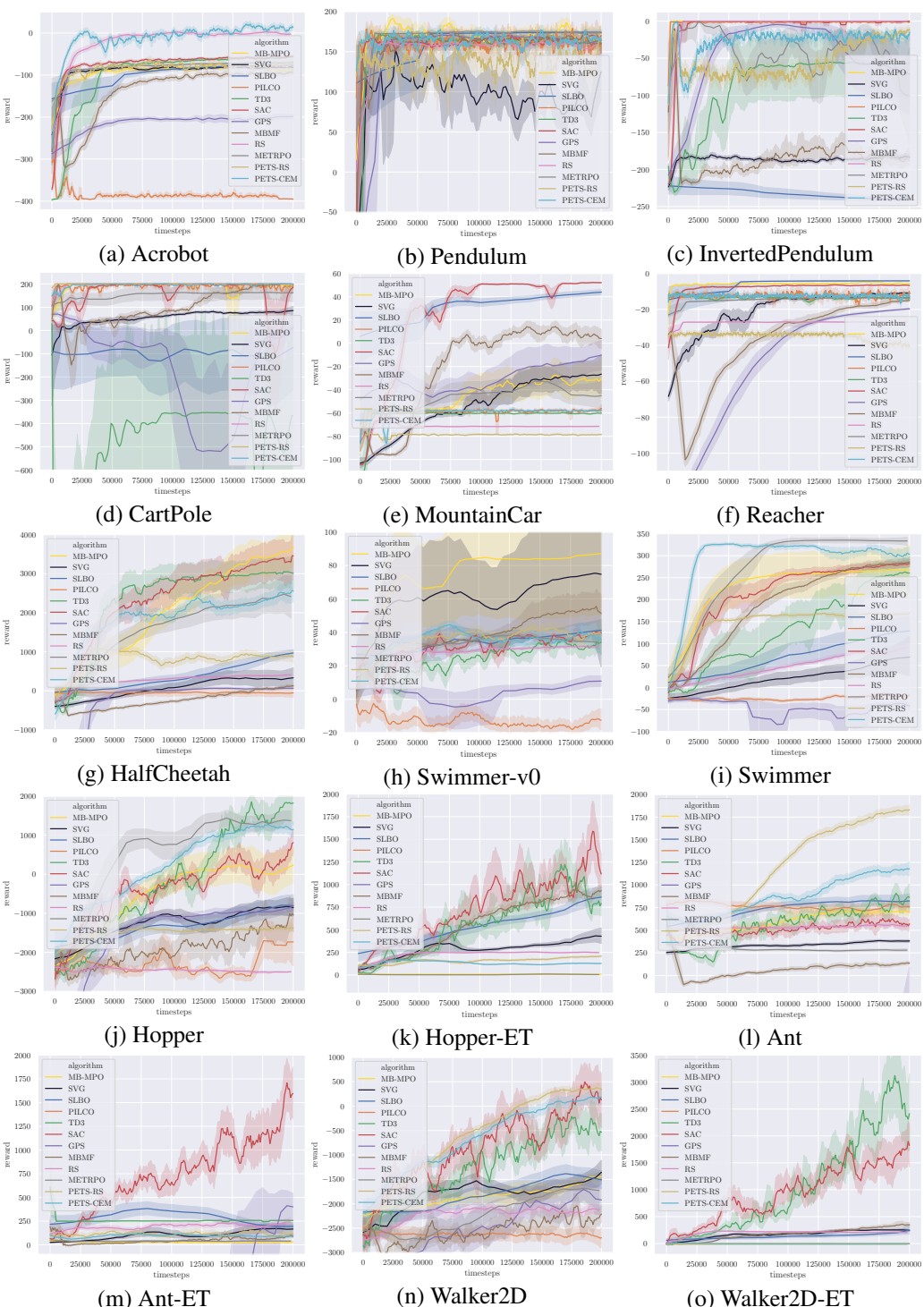

Figure 5: Performance curve for MBRL algorithms. There are still 3 more figures in a continued Figure 6.

In this appendix section, we include all the curves of every algorithms in Figure 5 and Figure 6. Some of the GPS curves and PILCO curves are not shown in the figures. We note that this is because their reward scale is sometimes very different from other algorithms.

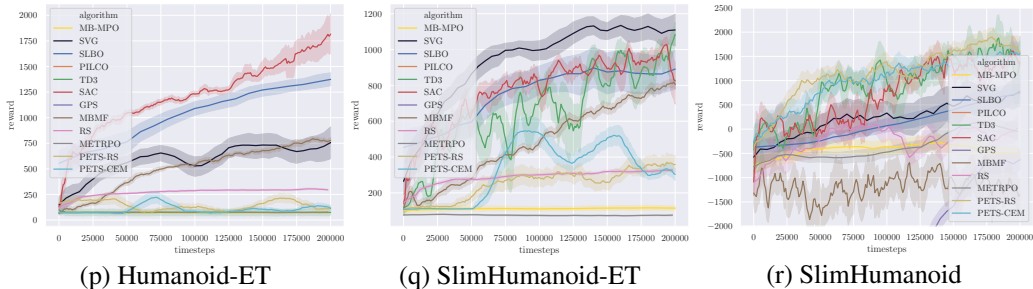

(p) Humanoid-ET      (q) SlimHumanoid-ET      (r) SlimHumanoid

Figure 6: (Continued) Performance curve for MBRL algorithms.

| | RS | MBMF | PETS | PETS-RS | METRPO | GPS | PILCO | SVG | MB-MPO | SLBO | PPO | TRPO | TD3 | SAC |
|---|---|---|---|---|---|---|---|---|---|---|---|---|---|---|
| Average | 10.23 | 4.41 | 7.47 | 4.66 | 4.01 | 6.68 | 120 | 1.42 | 44.68 | 5.32 | 0.04 | 0.031 | 3.87 | 3.04 |
| Standard deviation | 2.16 | 0.74 | 3.96 | 1.45 | 1.40 | 7.22 | N. A. | 0.33 | 9.95 | 1.63 | 0.0172 | 0.0116 | 0.89 | 0.67 |

Table 18: Wall-clock time in hours averaged for each algorithm trained for 200k time-steps from the original Table 2.

In addition to Table 4, we also have the performance of SVG trained for 1 million time-steps. For the environments of HalfCheetah, Ant, Walker2D and Hopper, the performance is respectively $578.7 \pm 239.1$, $472.0 \pm 48.1$, $-1168.2 \pm 537.2$ and $-753.4 \pm 580.8$.

# D   NOISY ENVIRONMENTS

In this appendix section, we provide more details of the performance with noise for each algorithm. In Figure 7 and Figure 8, we show the curves of different algorithms, and in Table 19 and Table 20 we show the performance numbers at the end the of training. The pink color indicates a decrease of performance, while the green color indicates a increase of performance, and black color indicates a almost the same performance.

| | Cheetah | Cheetah, $\sigma_o = 0.1$ | Cheetah, $\sigma_o = 0.01$ | Cheetah, $\sigma_a = 0.1$ | Cheetah, $\sigma_a = 0.03$ |
|---|---|---|---|---|---|
| iLQR | $2142.6 \pm 137.7$ | $-25.3 \pm 127.5$ | $187.2 \pm 102.9$ | $261.2 \pm 106.8$ | $310.1 \pm 112.6$ |
| GT-PETS | $14777.2 \pm 13964.2$ | $1638.5 \pm 188.5$ | $9226.5 \pm 8893.4$ | $11484.5 \pm 12264.7$ | $13160.6 \pm 13642.6$ |
| GT-RS | $815.7 \pm 38.5$ | $6.6 \pm 52.5$ | $493.3 \pm 38.3$ | $604.8 \pm 42.7$ | $645.7 \pm 39.6$ |
| RS | $421.0 \pm 55.2$ | $146.2 \pm 19.9$ | $423.1 \pm 28.7$ | $445.8 \pm 19.2$ | $442.3 \pm 26.0$ |
| MB-MF | $126.9 \pm 72.7$ | $146.1 \pm 87.8$ | $232.1 \pm 122.0$ | $184.0 \pm 148.9$ | $257.0 \pm 96.6$ |
| PETS | $2795.3 \pm 879.9$ | $1879.5 \pm 801.5$ | $2410.3 \pm 844.0$ | $2427.5 \pm 674.1$ | $2427.2 \pm 1118.6$ |
| PETS-RS | $966.9 \pm 471.6$ | $217.0 \pm 193.4$ | $814.9 \pm 678.6$ | $1128.6 \pm 674.2$ | $1017.5 \pm 734.9$ |
| ME-TRPO | $2283.7 \pm 900.4$ | $409.4 \pm 834.2$ | $1396.9 \pm 834.8$ | $1319.8 \pm 698.0$ | $2122.9 \pm 889.1$ |
| GPS | $52.3 \pm 41.7$ | $-6.8 \pm 13.6$ | $175.2 \pm 169.4$ | $41.6 \pm 45.7$ | $94.0 \pm 57.0$ |
| PILCO | $-41.9 \pm 267.0$ | $-282.0 \pm 258.4$ | $-275.4 \pm 164.6$ | $-175.6 \pm 284.1$ | $-260.8 \pm 290.3$ |
| SVG | $336.6 \pm 387.6$ | $0.1 \pm 271.3$ | $240.8 \pm 236.6$ | $163.5 \pm 338.6$ | $21.9 \pm 81.0$ |
| MB-MPO | $3639.0 \pm 1185.8$ | $2356.4 \pm 734.4$ | $3635.5 \pm 1486.8$ | $3372.9 \pm 1373$ | $3718.7 \pm 922.3$ |
| SLBO | $1097.7 \pm 166.4$ | $212.5 \pm 279.6$ | $1244.8 \pm 604.0$ | $1593.2 \pm 265.0$ | $731.1 \pm 215.8$ |
| PPO | $17.2 \pm 84.4$ | $-113.3 \pm 92.8$ | $-83.1 \pm 117.7$ | $-28.0 \pm 54.1$ | $-35.5 \pm 87.8$ |
| TRPO | $-12.0 \pm 85.5$ | $-146.0 \pm 67.4$ | $9.4 \pm 57.6$ | $-32.7 \pm 110.9$ | $-70.9 \pm 71.9$ |
| TD3 | $3614.3 \pm 82.1$ | $895.7 \pm 61.6$ | $817.3 \pm 11.0$ | $4256.5 \pm 117.4$ | $3941.8 \pm 61.3$ |
| SAC | $4000.7 \pm 202.1$ | $1146.7 \pm 67.9$ | $3869.2 \pm 88.2$ | $3530.5 \pm 67.8$ | $3708.1 \pm 96.2$ |

Table 19: The performance of each algorithm in noisy HalfCheetah (referred to in short hand as "Cheetah") environments. The green and red colors indicate increase and decrease in performance, respectively.

# E   PLANNING HORIZON DILEMMA GRID SEARCH

We note that we also perform the dilemma search with different population size with learnt PETS-CEM. We experiment both with the HalfCheetah in our benchmarking environments, as well as the environments from Chua et al. (2018), whose observation is further pre-processed. It can be seen from the figure that, planning horizon dilemma exists with different population size. We also show that observation pre-processing can affect the performance by a large margin.

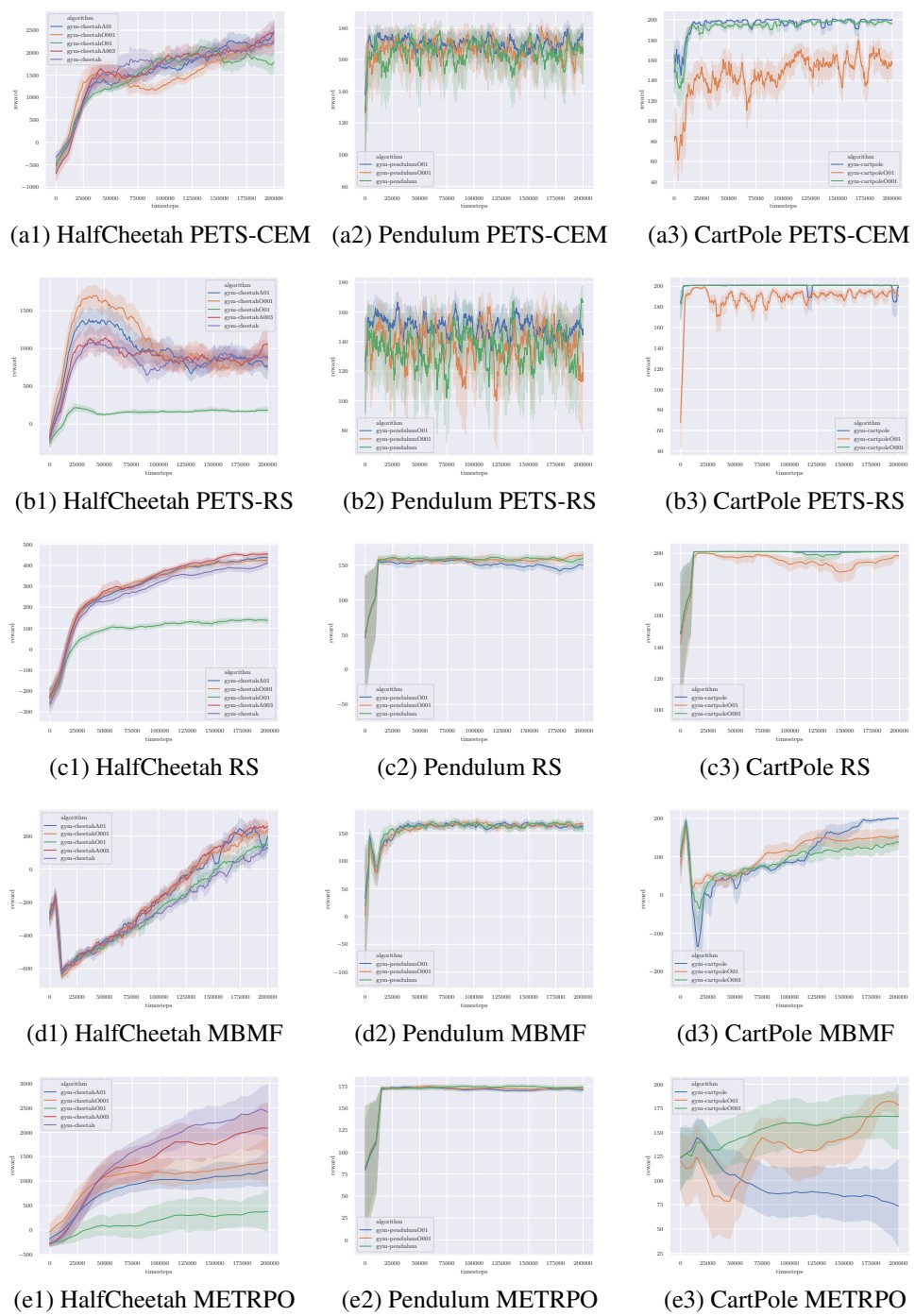

Figure 7: The performance curve for algorithms with noise. We represent the noise standard deviation with "O" and "A" respectively for the noise added to the observation and action space.

# F PLANNING HORIZON DILEMMA IN DYNA ALGORITHMS

In this section, we study how the environment length and imaginary environment length (or planning horizon) affect the performance. More specifically, we test with HalfCheetah and Ant, using different environment length form [100, 200, 500, 1000]. For the planning horizon, besides the matching

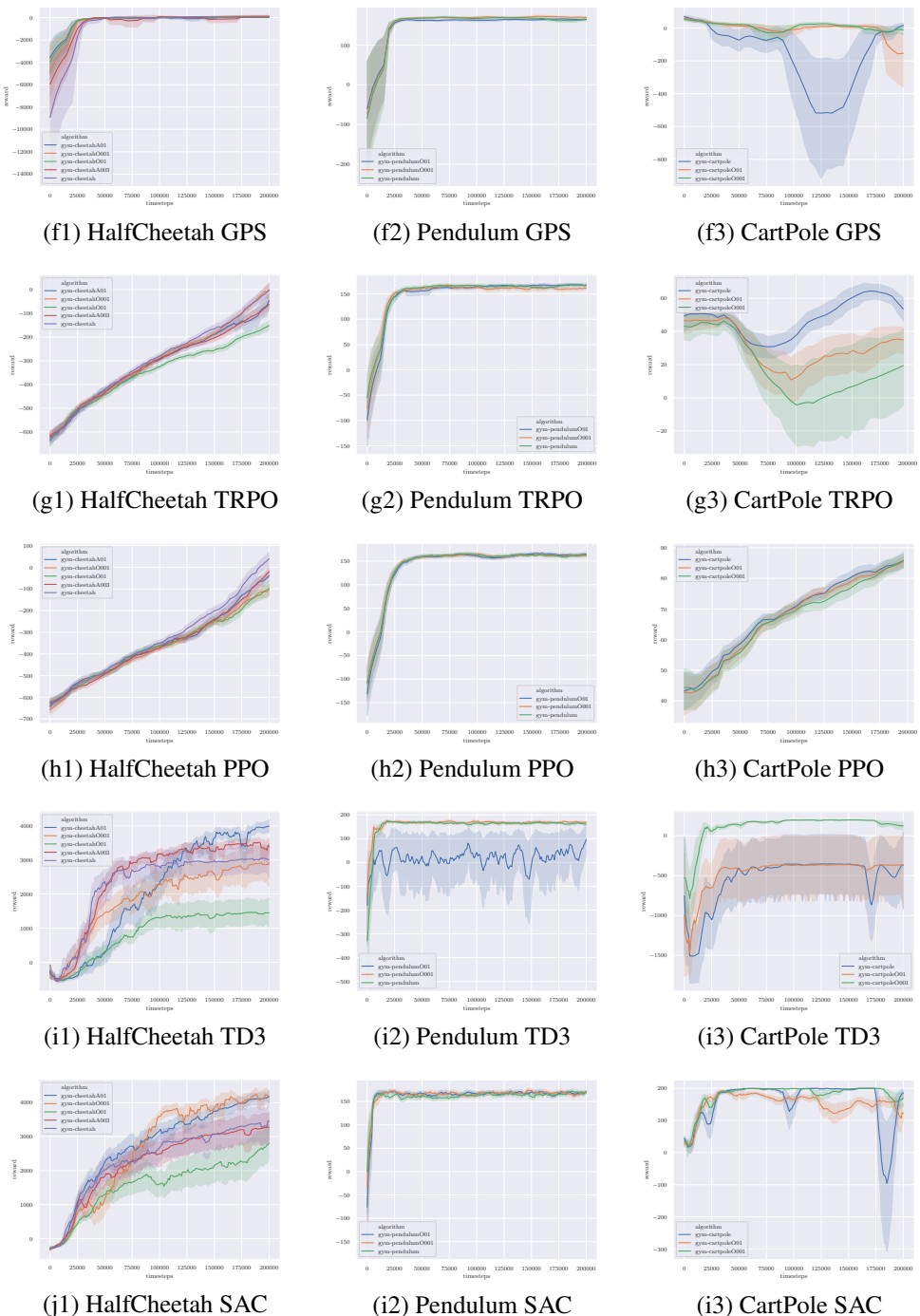

Figure 8: (Continued) The performance curve for algorithms with noise. We represent the noise standard deviation with "O" and "A" respectively for the noise added to the observation and action space.

length, we also test all the length from [100, 200, 500, 800, 1000]. The figures are shown in Figure 10, and the tables are shown in Table 21.

Note that we also include planning horizon longer than the actual environment length for reference. For example, for the Ant with 100 environment length, we also include results using 200, 500, 800,

| | Pendulum | Pendulum, $\sigma_o = 0.1$ | Pendulum, $\sigma_o = 0.01$ | Cart-Pole | Cart-Pole, $\sigma_o = 0.1$ | Cart-Pole, $\sigma_o = 0.01$ |
|---|---|---|---|---|---|---|
| iLQR | $160.8 \pm 29.8$ | $-357.9 \pm 251.9$ | $-2.2 \pm 166.5$ | $200.3 \pm 0.6$ | $197.8 \pm 2.9$ | $200.4 \pm 0.7$ |
| GT-PETS | $170.5 \pm 35.2$ | $171.4 \pm 26.2$ | $157.3 \pm 66.3$ | $200.9 \pm 0.1$ | $199.5 \pm 1.2$ | $200.9 \pm 0.1$ |
| GT-RS | $171.5 \pm 31.8$ | $125.2 \pm 40.3$ | $157.8 \pm 39.1$ | $201.0 \pm 0.0$ | $200.2 \pm 0.3$ | $201.0 \pm 0.0$ |
| RS | $164.4 \pm 9.1$ | $154.5 \pm 12.9$ | $160.1 \pm 6.7$ | $201.0 \pm 0.0$ | $197.7 \pm 4.5$ | $200.9 \pm 0.0$ |
| MB-MF | $157.5 \pm 13.2$ | $162.8 \pm 14.7$ | $165.9 \pm 8.5$ | $199.7 \pm 1.2$ | $152.3 \pm 48.3$ | $137.9 \pm 48.5$ |
| PETS | $167.4 \pm 53.0$ | $174.7 \pm 27.8$ | $166.7 \pm 52.0$ | $199.5 \pm 3.0$ | $156.6 \pm 50.3$ | $196.1 \pm 5.1$ |
| PETS-RS | $167.9 \pm 35.8$ | $148.0 \pm 58.6$ | $113.6 \pm 124.1$ | $195.0 \pm 28.0$ | $192.3 \pm 20.6$ | $200.8 \pm 0.2$ |
| ME-TRPO | $177.3 \pm 1.9$ | $173.3 \pm 3.2$ | $173.7 \pm 4.8$ | $160.1 \pm 69.1$ | $174.9 \pm 21.9$ | $165.9 \pm 58.5$ |
| GPS | $162.7 \pm 7.6$ | $162.2 \pm 4.5$ | $168.9 \pm 6.8$ | $14.4 \pm 18.6$ | $-479.8 \pm 859.7$ | $-22.7 \pm 53.8$ |
| PILCO | $-132.6 \pm 410.1$ | $-211.6 \pm 272.1$ | $168.9 \pm 30.5$ | $-1.9 \pm 155.9$ | $139.9 \pm 54.8$ | $-2060.1 \pm 14.9$ |
| SVG | $141.4 \pm 62.4$ | $86.7 \pm 34.6$ | $78.8 \pm 73.2$ | $82.1 \pm 31.9$ | $119.2 \pm 46.3$ | $106.6 \pm 42.0$ |
| MB-MPO | $171.2 \pm 26.9$ | $178.4 \pm 22.2$ | $183.8 \pm 19.9$ | $199.3 \pm 2.3$ | $-65.1 \pm 542.6$ | $198.2 \pm 1.8$ |
| SLBO | $173.5 \pm 2.5$ | $171.1 \pm 1.5$ | $173.6 \pm 2.4$ | $78.0 \pm 166.6$ | $-691.7 \pm 801.0$ | $-141.8 \pm 167.5$ |
| PPO | $163.4 \pm 8.0$ | $165.9 \pm 15.4$ | $157.3 \pm 12.6$ | $86.5 \pm 7.8$ | $120.5 \pm 42.9$ | $120.3 \pm 46.7$ |
| TRPO | $166.7 \pm 7.3$ | $167.5 \pm 6.7$ | $161.1 \pm 13.0$ | $47.3 \pm 15.7$ | $-572.3 \pm 368.0$ | $-818.0 \pm 288.1$ |
| TD3 | $161.4 \pm 14.4$ | $169.2 \pm 13.1$ | $170.2 \pm 7.2$ | $196.0 \pm 3.1$ | $190.4 \pm 4.7$ | $180.9 \pm 8.2$ |
| SAC | $168.2 \pm 9.5$ | $169.3 \pm 5.6$ | $169.1 \pm 12.6$ | $199.4 \pm 0.4$ | $60.9 \pm 23.4$ | $70.7 \pm 11.4$ |

Table 20: The performance of each algorithm in noisy Pendulum and Cart-Pole environments. The green and red colors indicate increase and decrease in performance, respectively. Numbers in black indicate no significant change compared to the default performance.

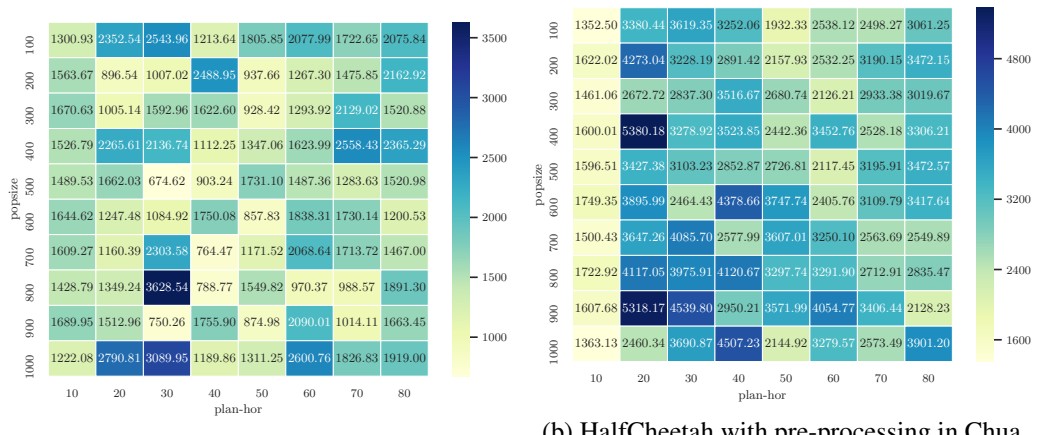

(a) HalfCheetah.

(b) HalfCheetah with pre-processing in Chua et al. (2018).

Figure 9: The performance grid using different planning horizon and depth.

| Environment | Original Length | Horizon=100 | Horizon=200 | Horizon=500 | Horizon=800 | Horizon=1000 |
|---|---|---|---|---|---|---|
| HalfCheetah | Env-100 | $250.7 \pm 32.1$ | $290.3 \pm 44.5$ | $222.0 \pm 34.1$ | $253.0 \pm 22.3$ | $243.7 \pm 41.7$ |
| HalfCheetah | Env-200 | $422.7 \pm 143.7$ | $675.4 \pm 139.6$ | $529.0 \pm 50.0$ | $451.4 \pm 124.5$ | $528.1 \pm 74.7$ |
| HalfCheetah | Env-500 | $816.6 \pm 466.0$ | $583.4 \pm 392.7$ | $399.2 \pm 250.5$ | $986.9 \pm 501.9$ | $1062.7 \pm 182.0$ |
| HalfCheetah | Env-1000 | $1312.1 \pm 656.1$ | $1514.2 \pm 1001.5$ | $1522.6 \pm 456.3$ | $1544.2 \pm 1349.0$ | $2027.5 \pm 1125.5$ |
| Ant | Env-100 | $1207.8 \pm 41.6$ | $1142.2 \pm 25.7$ | $1111.9 \pm 35.3$ | $1103.7 \pm 70.9$ | $1085.5 \pm 22.9$ |
| Ant | Env-200 | $1249.9 \pm 127.7$ | $1172.7 \pm 36.4$ | $1136.9 \pm 32.6$ | $1079.7 \pm 37.3$ | $1096.8 \pm 18.6$ |
| Ant | Env-500 | $1397.6 \pm 49.9$ | $1319.1 \pm 50.1$ | $1423.6 \pm 46.2$ | $1287.3 \pm 118.7$ | $1331.5 \pm 92.9$ |
| Ant | Env-1000 | $1666.2 \pm 201.9$ | $1646.0 \pm 151.8$ | $1680.7 \pm 255.3$ | $1530.7 \pm 48.0$ | $1647.2 \pm 118.5$ |

Table 21: The performance for different environment length and planning horizon in SLBO summarized in to a table. HalfCheetah and Ant were used in the experiments.

1000 planning horizon. As we can see, for the HalfCheetah environment, increasing planning horizon does not have obvious affects on the performance. In the Ant environments with different environment lengths, a planning horizon of 100 usually produces the best performance, instead of the longer ones.

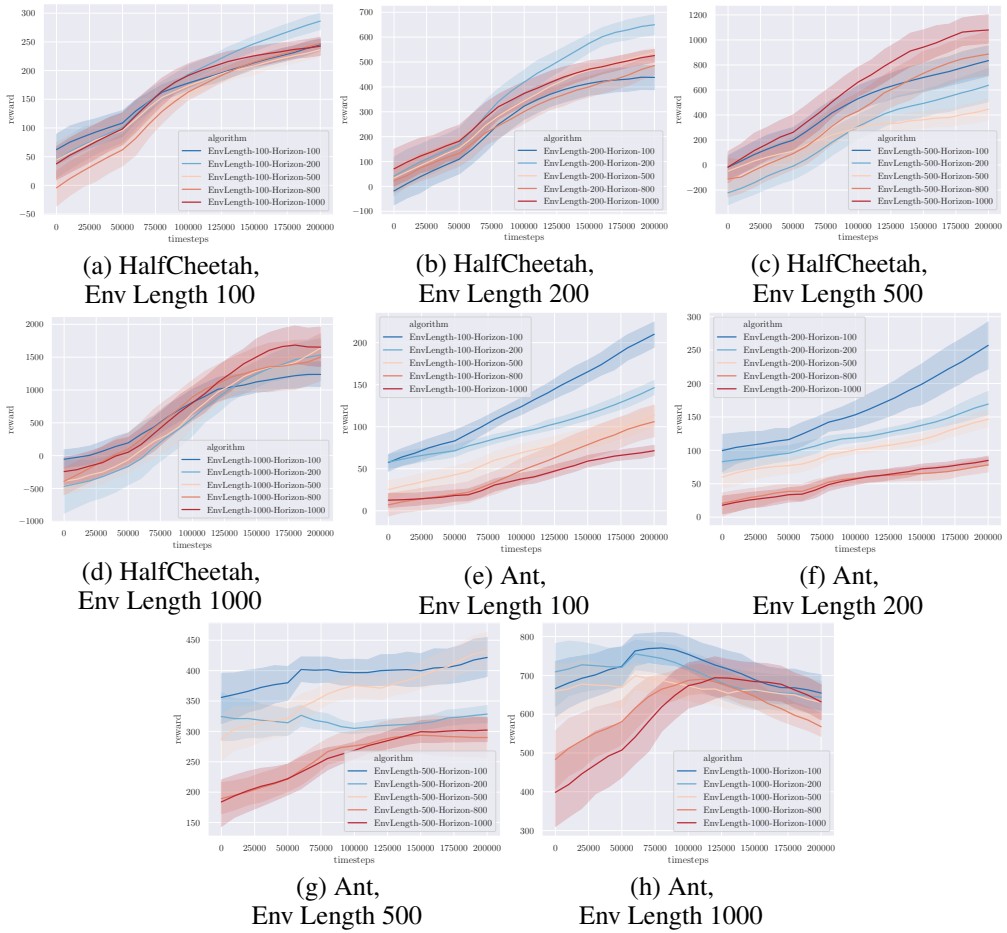

Figure 10: The performance curve for different environment length and planning horizon in SLBO. HalfCheetah and Ant were used in the experiments.

## G    DYNAMICS NETWORK STRUCTURE AND CAPACITY

We also study how the structure or the capacity of the dynamics network affect the performance. In Figure 11, we show the performance of PETS-CEM on HalfCheetah using different networks to learn the dynamics. In Rajeswaran et al. (2017), the authors show that two simple networks, linear network and RBF network, can be used as the policy network, which obtains similar performance compared with using multi-layer perceptron (MLP) in model-free reinforcement learning. Therefore, we use linear network and RBF network to learn the dynamics in model-based reinforcement learning. We also test the performance using wider and deeper networks.

As we can see from Figure 11, in model-based RL, linear network and RBF network lead to catastrophic performance drop, indicating multi-layer neural network (MLP) is needed to learn the features to model forward dynamics. On the other hand, increasing the capacity of the MLP by making it much deeper and wider does not seem to increase the performance either.

## H    NON-MUJOCO ENVIRONMENTS

To facilitate research, we also provide simulated agents that uses free physics engines other than MuJoCo. Some of the results can be shown in Figure 12. We note that these environments have different reward and observation functions from the environments benchmarked in the main paper. Therefore the results can not be used to compare with the results of the environments based on MuJoCo.

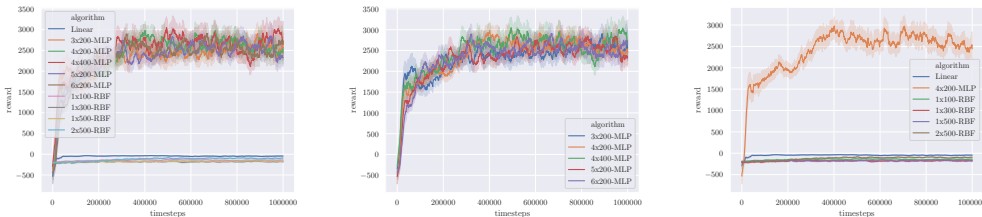

Figure 11: The performance of PETS-CEM on HalfCheetah using different network structures.

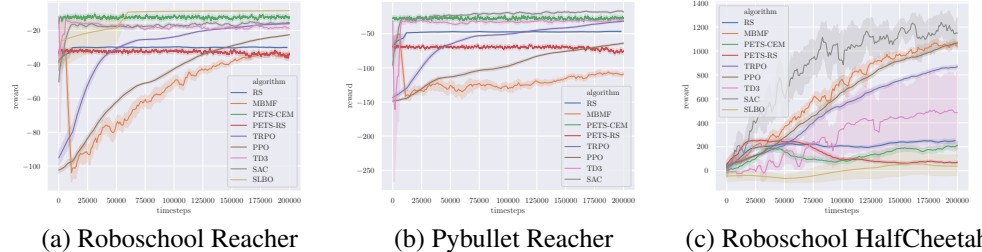

| (a) Roboschool Reacher | (b) Pybullet Reacher | (c) Roboschool HalfCheetah |
|---|---|---|

Figure 12: The performance curve for some of the tasks based on Roboschool or Pybullet Klimov & Schulman (2017); AMD (2014); Ellenberger (2018).

We use the best hyper-parameters for the corresponding MuJoCo environments, which indicates potential performance gain can be obtained with more careful hyper-parameter search.

## I  EARLY TERMINATION

|  | GT-CEM | GT-CEM-ET | GT-CEM-ET, $\tau = 100$ | learned-CEM | learned-CEM-ET |
|---|---|---|---|---|---|
| Ant | $12115.3 \pm 209.7$ | $8074.2 \pm 210.2$ | $4339.8 \pm 87.8$ | $1165.5 \pm 226.9$ | $162.6 \pm 142.1$ |
| Hopper | $3232.3 \pm 192.3$ | $260.5 \pm 12.2$ | $817.8 \pm 217.6$ | $1125.0 \pm 679.6$ | $801.9 \pm 194.9$ |
| Walker2D | $7719.7 \pm 486.7$ | $105.3 \pm 36.6$ | $6310.3 \pm 55.0$ | $-493.0 \pm 583.7$ | $290.6 \pm 113.4$ |

Table 22: The performance of PETS algorithm with and without early termination.

In this appendix section, we include the results of several schemes we experiment with early termination. The early termination dilemma is universal in all MBRL algorithms we tested, including Dyna-algorithms, shooting algorithms, and algorithm that performs policy search with back-propagation through time. To study the problem, we majorly start with exploring shooting algorithms including RS, PETS-RS and PETS-CEM, which only relates to early termination during planning. In Table 23 and Table 24, we also include the results that the agent does not consider being terminated in planning, even if it will be terminated, which we represent as "Unaware".

|  | GT-CEM | GT-CEM+ET-Unaware | GT-CEM-ET | GT-CEM-ET, $\tau = 100$ |
|---|---|---|---|---|
| Ant | $12115.3 \pm 209.7$ | $226.0 \pm 178.6$ | $8074.2 \pm 210.2$ | $4339.8 \pm 87.8$ |
| Hopper | $3232.3 \pm 192.3$ | $256.8 \pm 16.3$ | $260.5 \pm 12.2$ | $817.8 \pm 217.6$ |
| Walker2D | $7719.7 \pm 486.7$ | $254.8 \pm 233.4$ | $105.3 \pm 36.6$ | $6310.3 \pm 55.0$ |

Table 23: The performance using ground-truth dynamics for CEM.

For the algorithms with unknown dynamics, we specifically study PETS. We design the following schemes.

**Scheme A**: The episode will not be terminated and the agent does not consider being terminated during planning.

|          | GT-RS | GT-RS-ET-Unaware | GT-RS-ET | GT-RS-ET, $\tau = 100$ |
|----------|-------|------------------|----------|------------------------|
| Ant      | $2709.1 \pm 631.1$ | $2519.0 \pm 469.8$ | $2083.8 \pm 537.2$ | $2083.8 \pm 537.2$ |
| Hopper   | $-2467.2 \pm 55.4$ | $209.5 \pm 46.8$ | $220.4 \pm 54.9$ | $289.8 \pm 30.5$ |
| Walker2D | $-1641.4 \pm 137.6$ | $207.9 \pm 27.2$ | $231.0 \pm 32.4$ | $258.3 \pm 51.5$ |

Table 24: The performance using ground-truth dynamics for RS.

|          | Scheme A | Scheme B | Scheme D | Scheme C | Scheme E |
|----------|----------|----------|----------|----------|----------|
| Ant      | $1165.5 \pm 226.9$ | $81.6 \pm 145.8$ | $171.0 \pm 177.3$ | $110.8 \pm 171.8$ | $162.6 \pm 142.1$ |
| Hopper   | $1125.0 \pm 679.6$ | $129.3 \pm 36.0$ | $701.7 \pm 173.6$ | $801.9 \pm 194.9$ | $684.1 \pm 157.2$ |
| Walker2D | $-493.0 \pm 583.7$ | $-2.5 \pm 6.8$ | $-79.1 \pm 172.4$ | $290.6 \pm 113.4$ | $142.8 \pm 150.6$ |

Table 25: The performance of PETS-CEM using learned dynamics at 200k time-steps.

|            | Ant-ET-Unaware | Ant-ET | Ant-ET-2xPenalty | Ant-ET-5xPenalty | Ant-ET-10xPenalty | Ant-ET-20xPenalty | Ant-ET-30Penalty |
|------------|----------------|--------|------------------|------------------|-------------------|-------------------|------------------|
| GT-CEM     | $226.0 \pm 178.6$ | $8074.2 \pm 210.2$ | $1940.9 \pm 2051.9$ | $8092.3 \pm 183.1$ | $7968.8 \pm 179.6$ | $7969.9 \pm 181.5$ | $7601.5 \pm 1140.8$ |
| GT-RS      | $2519.0 \pm 469.8$ | $2083.8 \pm 537.2$ | $2474.3 \pm 636.4$ | $2591.1 \pm 447.5$ | $2541.1 \pm 827.9$ | $2715.6 \pm 763.2$ | $2728.8 \pm 855.5$ |
| Learnt-PETS | $1165.5 \pm 226.9$ | $81.6 \pm 145.8$ | $196.4 \pm 176.7$ | $181.0 \pm 142.8$ | $205.5 \pm 186.0$ | $204.6 \pm 202.6$ | $188.3 \pm 130.7$ |

Table 26: The performance of agents using different alive bonus or depth penalty during planning.

**Scheme B**: The episode will be terminated early and the agent adds penalty in planning to avoid being terminated.

**Scheme C**: The episode will be terminated, and the agent pads zero rewards after the episode is terminated during planning.

**Scheme D**: The same as Scheme A except for that the episode will be terminated.

**Scheme E**: The same as Scheme C except for that agent is allow to interact with the environment for extra time-steps (100 time-steps for example) to learn dynamics around termination boundary.

The results are summarized in Table 25. We also study adding more alive bonus, i. e. more death penalty during planning, whose results are shown in Table 26.

