# OpenReview forum: "Benchmarking Model-Based Reinforcement Learning"
_ICLR.cc/2020/Conference — Reject_

### Official Review · AnonReviewer1 · 2019-10-21
**Official Blind Review #1**

**Rating:** 1

**Review:**

This paper performs an empirical comparison of a number of model-based RL (MBRL) algorithms over 18 benchmarking environments. The authors propose a set of common challenges for MBRL algorithms.

I appreciate the effort put into this evaluation and I do think it helps the community gain a better understanding of these types of algorithms. My main issue with the paper is that I don't find the evaluation thorough enough (for instance, no tabular environments are evaluated) and the writing still needs quite a bit of work. I encourage the authors to continue this line of work and improve on what they have for a future submission!

More detailed comments:
- Intro: "model-free algorithms ... high sample complexity limits largely their application to simulated domains." I'm not sure this is a fair criticism. Model-based are also mostly run on simulations, so sample efficiency is not necessarily the cause model-free are only run on simulations. Further, this statement kind of goes against your paper, since all your evaluations are on simulations!
- Intro: "2) Planning horizon dilemma: ... increasing planning horizon... can result in performance drops due to the curse of dimensionality..." and "similar performance gain[sic] are not yet observed in MBRL algorithms, which limits their effectiveness in complex environments" goes against the point you made earlier on in the intro about sample efficiency.
- Preliminaries: "In stochastic settings, it is common to represent the dynamics with a Gaussian distribution", this is only for continuous states. It would be nice if you could evaluate tabular environments as well.
- Sec 4.1: "we modify the reward funciton so that the gradient... exists..." which environments were modified and how did they have to be modified?
- Sec 4.1: You discuss early termination but have not defined what exactly you mean by it.
- Fig 1: 12 curves is still a lot and really hard to make much sense of. A lot of the colors are very similar.
- Sec 4.1: "it takes an impractically long time to train for 1 million time-steps for some of the MBRL algorithms" why?
- Table 1 is a *lot* of numbers and colors and is really hard to make much sense of. There are also so many acronyms on the LHS it's difficult to keep track.
- Table 2: What about memory usage?
- Sec 4.4: "Due to the limited exploration in baseline, the performance is sometimes increased after adding noise that encourages exploration." Why does this noise not help exploration in the baselines?
- Sec 4.5: "This points out that when learning models more data does not result in better performance." This seems like it's closely correlated with the particular form chosen for the model parameterization more than anything.
- Fig 3: y-axis says "Relative performance", relative to what?
- Sec 4.7: "Early termination...is a standard technique... to prevent the agent from visiting unpromising states or damaging states for real robots." I've never seen this used as a justification for this.
- Sec 4.7: "Table 1, indicates that early termination does in fact decrease the performance for MBRL algorithms", and then you say "We argue that to perform efficient learning in complex environments... early termination is almost necessary." Those two statements contradict each other.

Minor comments to improve writing:
- When using citations as nouns, use \citep so you get "success in areas including robotics (Lillicrap et al., 2015)" as opposed to "success in areas including robotics Lillicrap et al. (2015)" (the latter is what you have all throughout the paper).
- Sec 3.1: s/This class of algorithms learn policies/These class of algorithms learn policies/

**Experience Assessment:**

I have published in this field for several years.

**Review Assessment: Checking Correctness Of Derivations And Theory:**

N/A

**Review Assessment: Checking Correctness Of Experiments:**

I carefully checked the experiments.

**Review Assessment: Thoroughness In Paper Reading:**

I read the paper thoroughly.

---

> ### Author Response · Authors · 2019-11-15
> **Thank you for your review - we hope to have addressed your main concerns with the current version**
>
> We thank the reviewer for the suggestions, but we are very astonishing to find that reviewer gave a clear reject to this paper.
> We address each question in this response.
> Your review is crucial to this project, and we sincerely hope that after we address the questions, the reviewer will reconsider the scores.
>
> - Q1: No tabular environments are evaluated.
>
> Our project has a focus of continuous control problems, and it is not very common to include tabular environments in current benchmarks [5, 6, 7].
> One of the reasons is that many algorithms (for example, iLQG) are not designed to solve tabular environments.
> Therefore we cannot include them unless we mask them as continuous environments. But then again, they are not tabular anymore, and they are substantially easier than the continuous environments currently in the benchmark.
>
> Our benchmarks include environments of different level of difficulties, from environments such as Cart-pole and Pendulum, which can be efficiently solved within 5 episodes, to Humanoid, which requires millions of time-steps to solve.
> We believe our benchmark has the biggest difficulty spectrum to the best of our knowledge.
>
> - Q2: Inaccurate choice of words and ambiguous sentences.
>
> We modified the sentences in the paper to be more accurate. Please refer to the newest revision and general response for more details.
>
> - Q3: Sec 4.1: "we modify the reward function so that the gradient... exists..." which environments were modified and how did they have to be modified?
>
> We are sorry for the confusion. We refer the reviewer to Appendix A.1 for details.
>
> - Q4: Sec 4.1: "it takes an impractically long time to train for 1 million time-steps for some of the MBRL algorithms" why?
>
> Some of the algorithms can require an extremely large amount of time and computational resources to train (up to 600 hours for PILCO with one 6GB GPU).
> We also refer the readers to Table 2 for the wall-clock time in hours for details.
>
> - Q5: Sec 4.4: "Due to the limited exploration in baseline, the performance is sometimes increased after adding noise that encourages exploration." Why does this noise not help exploration in the baselines?
>
> We refer the reviewers to Table 3 for details. We believe there is a misunderstanding of the results. It does also help the baselines quite often.
>
> - Q6: Sec 4.5: "This points out that when learning models more data does not result in better performance." This seems like it's closely correlated with the particular form chosen for the model parameterization more than anything.
>
> It is indeed a very good question. We therefore include experiments with networks of different sizes and layers. Besides multilayer perceptron network We also include experiments using linear networks and RBF networks.
> We refer to Appendix G for details.
>
> - Q7: I have never seen "Early termination is a standard technique... to prevent the agent from visiting unpromising states or damaging states for real robots."
>
> We apologize for the confusion. We add the citations and refer the readers to [1,2,3,4,5] for details.
>
> - Q8: "Table 1, indicates that early termination decrease the performance", and then you say "in complex environments early termination is necessary." Those two statements contradict each other.
>
> Sorry for the confusion in the paper. They don’t necessarily contradict each other.
> Early termination (ET) is a common and important component in MFRL, which enables MFRL to solve many for high dimensional tasks. We refer the reviewer to [1,2,3,4], where it’s shown that removing ET will cause the training to fail.
> MBRL cannot solve the discussed tasks in the paper. Therefore we would hope ET could help the training. However, early termination instead hurts the MBRL’s performance.
> A future direction is therefore how to combines ET with MBRL algorithms.
>
> - Q9: Table 2: What about memory usage?
>
> We are sorry for not including the results in the earlier version. Here as a summary, most of the algorithms are not memory hungry. And we attached the table in the appendix response.
> The full table will be updated in the newest revision.
>
>
> We sincerely hope, after we have the valuable discussions and the questions are now addressed, the reviewer would recommend acceptance for our paper.

---

> > ### Author Response · Authors · 2019-11-15
> > **Appendix**
> >
> >                           Memory usage Table
> >
> >                 |  Cartpole    |    Cheetah      |       Humanoid    |
> > RS            | 4800 / 0      |  6400  / 0        |   7360  /  0          |
> > MBMF     | 4960 / 0      | 6720  /  0        |   7520  /   0         |
> > PETS        | 896  / 522   |  928   / 522     |   944  /  618        |
> > PETS-RS  | 896  / 554   |  912   / 553     |   960  /  746        |
> > METRPO  |  992  / 2009 | 1008  / 2010  |   1600  /  2009    |
> > PILCO      | 2304 / 167   |  N. A. /  N. A. |    N. A. /   N. A.  |
> > SVG         | 1168 / 5807 |  2368  / 5808  |   2560  /  6000   |
> > SLBO       | 1300 / 0       | 1600 /  0         |   2080 /   0         |
> > PPO         | 1120  / 0      | 1360  /   0       |   1400  /  0         |
> > TRPO       | 1408 / 0      | 1768  /  0        |    2072  /  0        |
> > TD3          | 128   / 0      | 256  /  0          |    384  /  0          |
> > SAC         | 112   / 0       | 256  /  0         |     352  /  0          |
> > Memory usage table, shown in the format of (RAM usage / GPU memory usage)
> >
> > [6] Tassa, Yuval, Yotam Doron, Alistair Muldal, Tom Erez, Yazhe Li, Diego de Las Casas, David Budden et al. "Deepmind control suite." arXiv preprint arXiv:1801.00690 (2018).
> > [7] Duan, Yan, Xi Chen, Rein Houthooft, John Schulman, and Pieter Abbeel. "Benchmarking deep reinforcement learning for continuous control." In International Conference on Machine Learning, pp. 1329-1338. 2016.
> > [8] Mania, Horia, Aurelia Guy, and Benjamin Recht. "Simple random search of static linear policies is competitive for reinforcement learning." Advances in Neural Information Processing Systems. 2018.

---

### Official Review · AnonReviewer3 · 2019-10-22
**Official Blind Review #3**

**Rating:** 6

**Review:**

This paper presents a systematic empirical evaluation of model-based RL algorithms on (mostly) continuous control environments from OpenAI Gym, with comparison to popular model-free algorithms. It identifies three challenges for model-based RL, learning the dynamics, selecting the planning horizon, and applying early termination to guide learning.

A systematic comparison of model-based RL algorithms is missing from the literature, and I believe that this paper does a fairly thorough job of providing such a comparison. A wide range of algorithms are selected, and the environments are representative of those commonly used in the literature. The first two challenges identified have been recognized in the literature. For example, Vemula et al. (2019) [1] discuss the planning horizon in random search RL algorithms.

However, I would like to see some results on the policy search algorithms such as PILCO in Section 4.5, even if they are on the simpler environments. Currently they are not represented in Table 4.

Minor comment:
1. There are several instances where the writing should be clarified, e.g. acronyms are not explained before they are used. For example, it would be helpful to the reader to define GT-CEM and GT-RS in Section 3, especially as Table 1 (page 6) comes before the text discussing those two algorithms (page 7).
2. Table 1 is a bit difficult to parse. Maybe it could be split up, or some algorithms/environments deferred to the appendix.

[1] Vemula, Anirudh, Wen Sun, and J. Bagnell. "Contrasting Exploration in Parameter and Action Space: A Zeroth-Order Optimization Perspective." The 22nd International Conference on Artificial Intelligence and Statistics. 2019.


**Experience Assessment:**

I have read many papers in this area.

**Review Assessment: Checking Correctness Of Derivations And Theory:**

N/A

**Review Assessment: Checking Correctness Of Experiments:**

I assessed the sensibility of the experiments.

**Review Assessment: Thoroughness In Paper Reading:**

I read the paper at least twice and used my best judgement in assessing the paper.

---

> ### Author Response · Authors · 2019-11-15
> **Response to Reviewer 3**
>
>
> We would like to apologize for the writing of the paper. We updated the paper and hope it has increased its clarity.
> We thank the reviewer for the acknowledgement and suggestions for the project, and we address the following questions.
>
> - Q1: The first two challenges identified have been recognized in the literature. For example, Vemula et al. (2019) [1] discuss the planning horizon in random search RL algorithms.
>
> We would like to thank the reviewer for the reference and we have updated them in the latest revision.
>
> - Q2: However, I would like to see some results on the policy search algorithms such as PILCO in Section 4.5, even if they are on the simpler environments. Currently they are not represented in Table 4.
>
> We didn’t include the performance of PILCO as PILCO is very expensive to train in terms of the time and computation resources (potentially 120 hours for 200k time-steps for 1 random seed).
> For simple environments, PILCO can usually solve the task within a few episodes, but can not solve high dimensional problems regardless of the training efforts, which can also be shown in Appendix C.
> We updated the results for SVG in the appendix.
> For HalfCheetah, Ant, Walker2D and Hopper, the performances of SVG are respectively
> |      HalfCheetah     |          Ant           |        Walker2D         |        Hopper          |
> |     578.7±239.1       |    472.0±48.1     |    -1168.2±537.2      |    -753.4±580.8      |
>
> -  Q3: There are several instances where the writing should be clarified. -. Table 1 is a bit difficult to parse.
> We thank the reviewer for the suggestions and updated the paper accordingly.
> We fixed missing references and clarified the definition of acronyms before they are referred to.
>
>
> We hope that our response has addressed the main concerns, and we also refer to general response for other updates.
>
> [1] Vemula, Anirudh, Wen Sun, and J. Bagnell. "Contrasting Exploration in Parameter and Action Space: A Zeroth-Order Optimization Perspective." The 22nd International Conference on Artificial Intelligence and Statistics. 2019.

---

### Official Review · AnonReviewer2 · 2019-10-23
**Official Blind Review #2**

**Rating:** 6

**Review:**

Summary of the paper
The authors benchmark 11 model-based and 4 model-free RL algorithms on 18 environments, based on OpenAI Gym.
The performance in each case is averaged across 4 different seeds.  Computation time is also reported.
Furthermore, the authors analyze the performance hit incurred from adding noise to the observations and to the actions
Finally, the authors propose to characterize what hinders the performance of model-based methods, ie., what they call the dynamics bottlenecks, the planning horizon dilemma, and the early termination dilemma.
As a conclusion, it turns out no clear winner stands out, which motivates further development of model-based approaches.

Strong and weak points
This is a very interesting empirical study, especially since
- it includes a comparison with model-free algorithms,
- it considers computational aspects and indicates what algorithms can be run in real-time,
- the authors use open-source software (PyBullet) as simulators, which makes the study more reproducible (although the code has not been shared yet)

But,
- 4 seeds averaged across is clearly low, given the well-known variance of RL algorithms. In fact, the std values in the tables prove this point. I understand the benchmark is heavy on computation, however, this would only have delayed the output of the numbers without requiring more work (and admittedly, been even more harmful for the ecology...).
- Not as a criticism but rather a suggestion, it would have been useful to summarize the table 2 by comparing the algorithms using normalized values (mean = 0, std = 1) averaged across the environments.
- One the (strongest) weak points for me remains the assumption of the given differentiable reward function, as learning the reward function might be challenging for the model, for instance when it is sparse.
- It is also surprising that the authors did not benchmark against nor cite (Ha, David, and Jürgen Schmidhuber. “Recurrent world models facilitate policy evolution.” NeurIPS 2018), especially since the code is open-source.
- I would have made the same remark for (Learning Latent Dynamics for Planning from Pixels,  Hafner et al.) but the paper does not seem to be not peer-reviewed. Edit: the paper is indeed published at ICML 2019. So the remark holds.

Question:
- could the author elaborate on the early termination? it is not precisely defined anywhere and yet, seems to be an important point.

Minor
- page 3, parentheses around Finn et al. (2017)
- page 3, ILQG, “is an model”

**Experience Assessment:**

I have published one or two papers in this area.

**Review Assessment: Checking Correctness Of Derivations And Theory:**

N/A

**Review Assessment: Checking Correctness Of Experiments:**

I carefully checked the experiments.

**Review Assessment: Thoroughness In Paper Reading:**

N/A

---

> ### Author Response · Authors · 2019-11-15
> **Response to Reviewer 2**
>
>
> We thank the reviewer for the acknowledgement and suggestions for the project. And we address the following questions.
>
> - Q1: 4 seeds averaged across is clearly low, given the well-known variance of RL algorithms. I understand the benchmark is heavy on computation, however, this would only have delayed the output of the numbers without requiring more work (and admittedly, been even more harmful for the ecology...).
>
> We thank again for your consideration and understanding! Ideally we hope we can run much more random seeds, but some of the algorithms take quite a long time to run and are expensive economically and ecologically. And while the std value is big, the mean values for many MBRL algorithms are relatively stable across different seeds, compared with MFRL algorithms.
> Therefore we decided to trade-off here and release the code for future researchers.
>
> - Q2: it would have been useful to summarize the table 2 by comparing the algorithms using normalized values (mean = 0, std = 1) averaged across the environments.
>
> We thank the reviewer for the advice. We now show the mean values and the std of different algorithms across environments in appendix C Table 18.
>
> - Q3: One (strongest) weak points for me remains the assumption of the given differential reward function, as learning the reward function might be challenging for the model, for instance when it is sparse.
>
> We are hoping we can summarize and evaluate the majority of existing MBRL algorithms by making this assumption.
> Some of the algorithms such as iLQG can only be evaluated under this assumption.
> We agree that tasks with sparse / unknown reward function is very important for the future of MBRL and but we leave it for future research.
>
> - Q4: Comparison and citation to worldmodel (David Ha, etc.) and PlaNet (Danijar etc.).
>
> We added citation and comparison of the two algorithms in the latest revision.
> Our benchmark consists of tasks with observation from the states. Both worldmodel and PlaNet are algorithms for tasks with images as observation, which are harder problems but essentially  they have the same RL formulation.
> Worldmodel can be regarded as a special case of Dyna algorithm, and PlaNet can be regarded as a special case of PETS-CEM algorithm, with both of them designing an elegant framework to learn representations from images. Therefore we didn’t benchmark them in the paper.

---

### Author Response · Authors · 2019-11-15
**General response**

We thank the reviewers for the valuable suggestions and updated the latest revision for better clarity and additional experiment results.

We address some of the common questions and summarize the updates here:

- 1: Please elaborate on the early termination.

Early termination in this project refers to terminating the episodes using prior knowledge to facilitate training.
For example, for the humanoid trained to run, we terminate it if it falls to the ground.
It is usually used in high dimensional locomotion tasks (OpenAI gym Humanoid for example), where the training becomes orders of magnitudes harder if we don’t early terminate the episodes.
It has been used in high dimensional locomotion tasks [1, 2, 3, 4, 5]. And we refer the reviewer to these literature for more information.
We apologize for not having defined the term, and updated the writing to formally introduce it in the latest revision.


- 2: More related work and comparisons
We thank the reviewers for their valuable suggestions for the missing related work. They are now updated accordingly in the latest revision.

- 3. Update experiment results
In the latest revision, we added more results. Some of the tables and figures provide better visualization and clarity. And some others were not released in the earlier version, but



[1] Peng, Xue Bin, Glen Berseth, and Michiel Van de Panne. "Terrain-adaptive locomotion skills using deep reinforcement learning." ACM Transactions on Graphics (TOG) 35.4 (2016): 81.
[2] Merel, J., Tassa, Y., Srinivasan, S., Lemmon, J., Wang, Z., Wayne, G., & Heess, N. (2017). Learning human behaviors from motion capture by adversarial imitation. arXiv preprint arXiv:1707.02201.
[3] Heess, Nicolas, Greg Wayne, Yuval Tassa, Timothy Lillicrap, Martin Riedmiller, and David Silver. "Learning and transfer of modulated locomotor controllers." arXiv preprint arXiv:1610.05182 (2016).
[4] Peng, X. B., Abbeel, P., Levine, S., & van de Panne, M. (2018). Deepmimic: Example-guided deep reinforcement learning of physics-based character skills. ACM Transactions on Graphics (TOG), 37(4), 143.
[5] Brockman, G., Cheung, V., Pettersson, L., Schneider, J., Schulman, J., Tang, J., & Zaremba, W. (2016). Openai gym. arXiv preprint arXiv:1606.01540.

---

### Public Comment · ~Harshit_Sikchi1 · 2020-03-12
**Differences of performance curve across previous literature**

I am trying to figure out the cause of differences between the results shown in this paper and the previous literature. For example, lets take the Figure 2. (a) HalfCheetah plots. The questions are:
1. Why does PETS-CEM perform so poorly here, when Figure 3. of PETS paper https://arxiv.org/pdf/1805.12114.pdf  show such nice performance on HalfCheetah.

2. Figure 1 of SAC paper https://arxiv.org/pdf/1812.05905.pdf also shows different performance. Also Figure 5 of TD3 paper https://arxiv.org/pdf/1802.09477.pdf show different performance on HalfCheetah.

I understand there is a lot of tuning and uncertainty in RL algorithms, but it would be helpful if you can give an explanation as to which factors contributed to these differences.

---

### Decision · Program_Chairs · 2019-12-19

**Decision:**

Reject

**Comment:**

This paper provide an extensive set of benchmarks for Deep Model-based RL algorithms.

This paper contains a large number of algorithms, environments, and empirical results. The reviewers all recognized the need for such a study to provide some clarity, insights, and common standards. The reviewers we concerned about several aspects of the implementation of the effort. (1) All the performance is based on 4 runs, smoothed curves, and default errors (often extensively overlapping). The paper cites Henderson et al, and yet does not follow the advice laid out therein. (2) The results were fairly inconclusive---perhaps to be expected---we didn't learn much (more on this below). (3) The paper has communication issues.

The overall approach taken was a bit perplexing. Some algorithms we given access to the dynamics. The reward functions were converted to diff. forms, and early stopping in a domain specific way was employed. This all seems like simplifying the problem in different ways so that some methods can be competitive, but it is not at all clear why. If we take the typical full rl problem and limit domain knowledge, many of these approaches cannot be applied and others will fail. Those are the results we want. One could actually view these choices are unfair to more general algorithms---algorithms that need diff rewards pay no price for this assumption. This also leads to funny things, for example, like using position as the reward in mountain car (totally non-standard, and invalid without discounting). The paper claims a method can solve MC, but that is unclear from the graph. The paper motivates the entire enterprise based on the claimed lack of standardization in the literature, but then proceeds to redefine classic control tasks with little discussion or explanation.

The paper has communication issues. For example, all the domains are use continuous actions (and the others in the response highlight that is their main focus), but this is never stated in the paper. The paper refers to and varies "environment length", but this was not defined in the paper and has no obvious meaning. The tasks are presumably discounted but the the value of gamma is not specified anywhere in the paper (could be there, but I searched for a while). Pages of parameter settings in the appendix with many not discussed or their ranges justified.

This paper is ambitious, but I urge the authors to perhaps limit the scope and do less, and consider a slightly broader audience in both the writing and experiment design.